



# Are hydrological pathways and variability in groundwater chemistry linked in the riparian boreal forest?

Stefan W. Ploum[1], Hjalmar Laudon[1], Andrés Peralta-Tapia[2], Lenka Kuglerová[1]

[1]Department of Forest Ecology and Management, Swedish University of Agricultural Sciences, 901 86 Umeå, Sweden
5  [2]Department of Ecology and Environmental Sciences, Umeå University, 901 87 Umeå, Sweden

*Correspondence to*: Stefan W. Ploum (stefan.ploum@slu.se)

**Abstract.**

10 The riparian zone, or near-stream area, plays a fundamental role for the biogeochemistry of headwaters. Here groundwater undergoes chemical transformation before it enters the stream. However, the riparian zone is not uniform and spatial variability of groundwater hydrology and chemistry can be large. Terrestrial topographic depressions create hydrological pathways towards focal points in the riparian zone, which we refer to as Discrete Riparian Inflow Points (DRIPs). Given the important chemical function of the riparian zone, we therefore ask the question: are hydrological pathways and chemical variability 15 linked in the riparian boreal forest? To answer this question, we sampled riparian groundwater during six campaigns across three boreal headwater streams in Sweden. The groundwater wells were distributed in DRIP and non-DRIP pairs (60 wells), following transects from upland (20 meters lateral distance) to near stream area (<5 meters lateral distance). The variability in dissolved organic carbon (DOC), pH and electrical conductivity (EC) was analyzed using linear mixed effect models (LMM). We explained the variability using three factors: distance from the stream, seasonality and hydrological 20 connection/groundwater condition. Our results showed that DRIPs provided DOC rich water (34 mg/l) with relatively low EC (36 µS/cm). The so-called 'non-DRIP' riparian water had on average 40% lower DOC concentrations (20 mg/l) and 45% higher EC (52 µS/cm). Moreover, DRIPs were chemically more stable from the upland area to the stream (20-25 meter) and more constant throughout different seasons. In contrary, non-DRIP water transformed distinctly in the last 25 meters to the stream, and chemical variability also changed across the seasons. We concluded that hydrological pathways and spatial 25 variability in groundwater are linked, and that DRIPs are important control points in the boreal landscape. This finding is important for upscaling of stream inputs in boreal ecosystems, and for implementing hydrological adaptation into riparian forest management. However, for understanding underlying processes and mechanisms, we propose to investigate spatial variability of groundwater chemistry in a non-binary context, focusing on how groundwater chemistry relate to a gradient of hydrological fluctuations, soil properties and landscape characteristics.

30



## 1 Introduction

Headwaters can be seen as the capillaries of the landscape. The rich variety in hydrology, biology and chemistry of headwaters is tightly connected to processes in the surrounding landscape (Bishop et al., 2008; Hunsaker and Levine, 1995). Newly introduced terrestrial water accounts for a large part of the streamflow, magnifying groundwater controls on stream processes

(Hotchkiss et al., 2015). These controls are governed by groundwater-surface water exchange in the last interface between the landscape and stream ecosystems (Hayashi and Rosenberry, 2002). This near-stream area, or riparian zone (RZ), holds important functions such as chemical transformation of hillslope water (Cirmo and McDonnell, 1997), thermal regulation (Davies-Colley and Rutherford, 2005) and erosion control (Smith, 1976). A few characteristics of the boreal RZ that leads to its unique ecosystem functions are high groundwater levels, dynamic redox potential, build-up of soil organic matter, and

diverse vegetation (Grabs et al., 2012; Kuglerová et al., 2014b; Lidman et al., 2017). In terms of the hydrological role of the RZ , it has been demonstrated that riparian water dominates streamflow generation, instead of event-based water contributions from hillslopes (McGlynn and McDonnell, 2003). Combined with the chemical transformation of water in the riparian zone, stream biogeochemistry is therefore largely controlled by riparian zones (Ledesma et al., 2018; Lidman et al., 2017). However, RZ's are not just homogenous strips surrounding surface waters, but contain an array of  heterogeneities in hydrogeology, soil

development and vegetation across small spatial scales (Buttle, 2002; Kuglerová et al., 2014b). It is therefore important to further investigate which parts of the riparian zone matter most for stream flow generation and associated water chemistry.

Saturated areas of the riparian zone provide the majority of stream water (Penna et al., 2016). Traditionally, streamflow generation has often been assumed to be driven by spatially diffuse groundwater exchange often released at a constant rate.

Although this assumption is convenient for hydrological modelling efforts, in practice the occurrence of spatially focused discharges of groundwater is more rule than exception (Briggs and Hare, 2018). Also from a chemical perspective, it is necessary to account for spatial dynamics within the RZ. For example, wet riparian areas have been associated with denitrification, and retention and transformation of (labile) OM, compared to drier, oxic, riparian soils (Blackburn et al., 2017; Burgin and Groffman, 2012; Ledesma et al., 2018). In terms of vegetation, groundwater discharge zones are hotspots for

diversity (Kuglerová et al., 2014a). Taken all together, contributions of such focused riparian inputs could therefore function as important *control points* in the landscape (Bernhardt et al., 2017). The difficulty is that incorporating these control points into models or practical applications means that they have to be characterized in order to explain stream dynamics. Especially for informing distributed models that overpass catchment scale, determination and characterization of these control points remains one of the challenges for the scientific community (Briggs and Hare, 2018).


The occurrence of saturated areas in the RZ is linked to preferential hydrological pathways that route upland water towards streams. Although subsurface pathways do not entirely follow surface topography (Devito et al., 2005), it has been demonstrated that topographic depressions are a good indicator for accumulation areas of water, ponding, shallow groundwater tables and concentrated flow paths in the near-stream area (Ågren et al., 2014; Jencso et al., 2009; Wallace et al., 2018). As

such, topographic models can predict where along a stream network disproportionally large amounts of groundwater connect with the stream (Leach et al., 2017). These so called discrete riparian inflow points (DRIPs), provide consistent flows of subsurface water during low flow periods, but have also been observed to be highly dynamic in their activation during hydrological events (Ploum et al., 2018). A recent study has showed increases in greenhouse gas concentrations in close downstream proximity of DRIPs, yet the magnitude of these increases varied temporally (Lupon et al., 2019). Also in Arctic

systems the presence of riparian wet areas partially explains stream $CO_2$ evasion (Rocher-Ros et al., 2019). These findings suggest that both the hydrological fluxes as well as biogeochemical reactions in the stream are associated with the hydrological activity of DRIPs. However, in order to determine whether DRIPs matter for stream biogeochemistry, chemical characterization of the discharging groundwater is needed.





Characterizing groundwater chemistry is an especially challenging task. Previously this challenge has been by-passed by assuming that groundwater is a well-mixed source of water (Kirchner, 2003), or by inferring groundwater chemistry from base flow chemistry of streams (Peralta-Tapia et al., 2015). However, even at the local scale spatial variability in groundwater

chemistry overrules temporal variation and requires regular sampling of extensive well networks (Kiewiet et al., 2019). Within meters of each other, groundwater signatures can vary greatly (Penna et al., 2016). Three key parameters for chemical characterization of groundwater are dissolved organic carbon (DOC), pH and ionic strength. DOC concentrations in groundwater is the result of interaction between water and carbon rich materials in the shallow subsurface environment that are associated with paludification (Lavoie et al., 2005). Apart from its role in food-web structures and carbon transport, DOC

also increases the acidity (decrease pH) of soils and surface waters (Buffam et al., 2007). Electrical conductivity (EC) can be used as a proxy for the ionic strength, or total amount of dissolved ions in water (Corwin and Lesch, 2005). Water contact time with minerals and weathering processes are important factors determining EC. Therefore, water that has long residence time with mineral soils typically has elevated EC levels.

In the context of spatial variability of riparian groundwater, it can be expected that DOC, pH and EC differ between DRIP and non-DRIP riparian areas (Fig. 1). DRIPs are associated with high groundwater levels and wet, organic rich soils with vegetation that favors wet conditions, while non-DRIPs have drier top soils and deeper groundwater levels (Kuglerová et al., 2014a). Inherent to their topographic setting, DRIPs drain a large upland area, while non-DRIPs typically drain only a small surrounding area of the riparian zone or they are recharge zones for adjacent DRIPs. Moreover, the water in DRIPs travels a

longer distance horizontally; in presumably wet, highly permeable, organic rich soil. Non-DRIP water, on the other hand, is likely to infiltrate through an oxic, organic rich top soil, before being transported a relative short distance horizontally through supposedly more mineral substrate. This implies that the contact time of the water with wet, organic soil and drier, mineral soil is different for both cases, which should lead to contrasting water chemistry. In this study we characterize groundwater in a paired well network that is specifically designed to incorporate saturated riparian areas (DRIPs) and drier parts of the riparian

zone (non-DRIPs). We hypothesize that groundwater in DRIPs has higher DOC concentrations and lower pH compared to non-DRIPs. The deeper groundwater levels in non-DRIP areas, and longer contact times with mineral soil relative to organic soil, leads us to expect that EC will be higher in non-DRIP water compared to DRIPs.

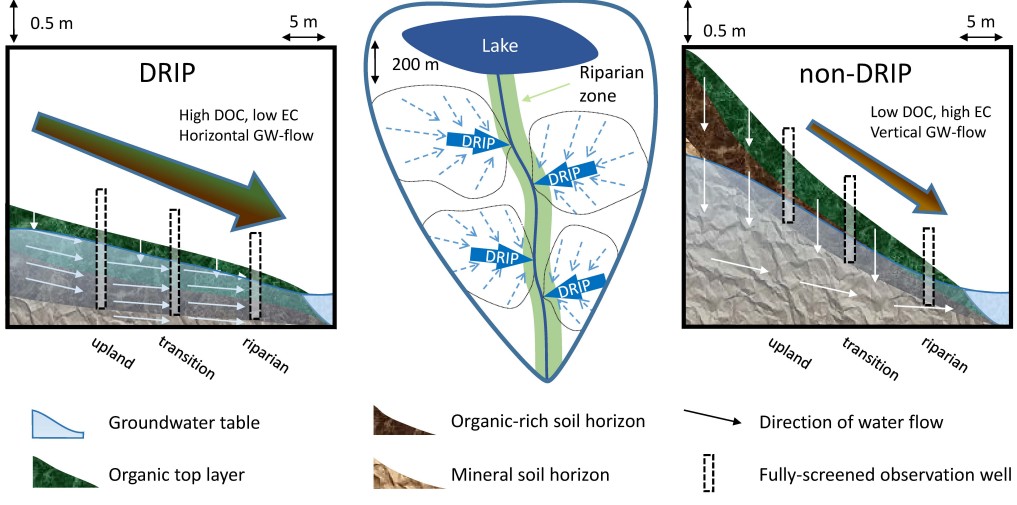

**Figure 1 Conceptual model of hydrological pathways in riparian boreal forest. Discrete Riparian Inflow Points (DRIPs) are focal**
**points in the riparian zone where pathways confluence before reaching the stream (central panel). Groundwater flow towards DRIP**



**and non-DRIP RZ (arrows in left and right panel) is conceptualized as predominantly horizontal and vertical, respectively. Green layers represent the approximate extent of the organic layer. Brown layers are riparian soils with high organic matter content. Light brown layers represent parent material or mineral horizons. Transparent blue overlay represents the groundwater table. Black bars represent well transects of respectively DRIP areas on the left-hand side and non-DRIP areas on the right-hand side. Large arrows**

**suggest relative hydrological contribution with color fill that matches soil layer with which groundwater has interacted most.**

## 2 Material and methods

To test our hypothesis we collected DRIP and non-DRIP groundwater across a riparian gradient during different seasons. Using linear mixed effect models (LMM's) we analyzed the role of DRIPs on biogeochemical composition of riparian groundwater, in relation to spatial and temporal variability. We performed our study in Krycklan, a boreal forested catchment

in northern Sweden.

### 2.1 Study area

The Krycklan catchment is situated near Vindeln, Sweden (64°14′N, 19°46′E, Fig. 2). The bedrock is predominately Svecofennian metasediments and metagreywacke. Quaternary deposits consist mostly of till (51%) and sorted sediments (30%). Land cover is dominated by forest (87%), and there is 9% mire cover. Furthermore there are sporadically thin soils and

rock, and a small fraction of arable land (2%). The climate is characterized as cold humid temperate type, with almost 6 months of snow cover. The yearly average temperature is 1.8 °C, and annual precipitation is 614 mm, and the annual mean runoff approximates 311 mm (Laudon et al., 2013). The streams along which the well network is situated are in the Svartberget research forest, referred to as C4, C6, and C8 (Laudon et al., 2013), with a drainage area of respectively 18, 110, and 230 ha. Catchments C4 and C6 have been widely studied in regard of lateral flow and groundwater and surface water interaction and

can be referred to in other studies as Kallkälsbäcken and Stortjärnbäcken (Laudon et al., 2004b, 2007). Flows vary from a few liters per second baseflow to 200 l/s peak flows. The yearly hydrograph is characterized by sustained baseflow throughout the winter months, followed by spring snowmelt floods in April and May. In summer and autumn low flow conditions are common with occasional rain-induced flow events. From November onwards, flow reduces as temperatures fall below 0 °C and baseflow conditions set in.



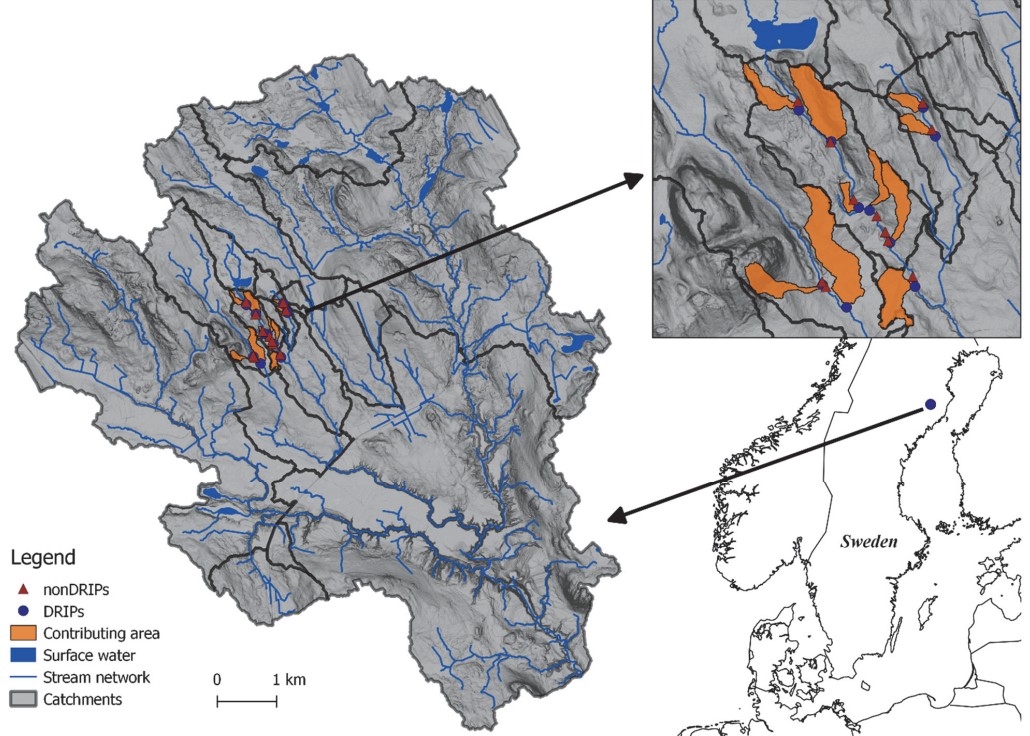

**Figure 2 Krycklan catchment in Northern Sweden. The upper right panel shows the particular study area where the well network has been installed. The red triangles and blue dots indicate respectively non-DRIP and DRIP transects, consisting of three wells placed at 20, 10 and <5 meters from the stream.**

### 2.2 DRIP definition

Saturated riparian areas have been defined in the landscape based on a topographic wetness index (2 ha threshold) and flow accumulation algorithms (Ågren et al., 2014; Beven and Kirkby, 1979; O'Callaghan and Mark, 1984). The selected wet areas (n = 10) have been field-validated and surveyed on species richness (Kuglerová et al., 2014a). For some sites chemical and thermal signatures further corroborated the location were riparian water discharged into the stream (Leach et al., 2017). These areas are referred to as discrete riparian inflow points, or DRIPs (Ploum et al., 2018). We use the term DRIPs for a collection of phenomena that have been described in literature using a variety of terms for the confluence of terrestrial water before it is incorporate in the stream network. Some of these existing terms are: groundwater discharge zone, groundwater hotspots, cryptic wetlands, swales, focused seepage, discrete seepage, springs, upwelling zones, preferential discharge, ephemeral streams and zero-order streams (Creed et al., 2003; Hayashi and Rosenberry, 2002; Tsuboyama et al., 2000). The existing terms often inherently refer to specific water sources (e.g. groundwater), specific morphology (e.g. stream) or process (e.g. upwelling), while in practice these confluences represent a spectrum of how hillslope water reaches surface water. Seeps, groundwater discharge, springs or similar terms are associated with relatively deep groundwater, however the water provided by DRIPs is not always groundwater, but can also be overland runoff consisting of rain or snowmelt water. In our case also the terminology referring to channels with temporary flow (ephemeral or intermittent streams, zero order channels) would not justify areas where we encounter inflow without a defined channel or flow path. With the DRIP term we indicate a confluence point in the riparian zone that provides a stream with terrestrial water, possibly provided by different sources of water over time.



### 2.3 Groundwater sampling and chemical analysis

The setup of this study consists of a well network with a total of 60 fully screened wells arranged in 10 paired transects. All wells were drilled until resistance, or an aquitard layer, which was in all cases within 1.5 meter from the soil surface. Each transect consisted of a riparian well, situated typically between 1 and 5 meter from the stream, a transition well at approximately

10 meters from the stream, and an upland well 20 meters from the stream. Transects followed the local topography, to approximate local hydraulic gradients and flow paths. The non-DRIP transects were installed close (<50 m) to each DRIP transect to ensure similarity in local conditions. Data collection involved six groundwater sampling campaigns. Water samples were collected during spring, summer and autumn of the 2016 and 2017 hydrological years. The well network was sampled using suction cup lysimeters and vacuumed glass bottles (Blackburn et al., 2017). The wells were pumped before installing the

suction cups. The bottles were collected after approximately 24 hours and subsampled, filtered and analyzed within 48 hours. In addition, a more intensive sampling campaign was conducted for a series of riparian wells only. These were sampled using a peristaltic pump. In total 359 samples were analyzed from the sampling campaigns, of which 200 from DRIP wells and 159 from non-DRIP wells. Non-DRIP wells occasionally had too low water level to collect a representative water sample. For analysis of dissolved organic carbon (DOC), a subsample was filtered (0.45 µm) into acid-washed high-density polyethylene

bottles (rinsed three times) and kept at 4 °C before laboratory analysis. DOC was measured by acidifying the sample and combustion using a Shimadzu TOC-V$_{PCH}$. The pH and EC were subsampled without headspace into acid-washed high-density polyethylene bottles (rinsed three times) and kept at 4 °C before laboratory analysis. Samples were analyzed using a Mettler Toledo DGi117-water probe for pH and Mettler Toledo InLab741 probe for electrical conductivity.

### 2.4 Statistical analysis

We used linear mixed-effect models (LMM) to analyze patterns in DOC, pH and EC. The analysis was performed in R using *lmer* models from the R-package *lme4* (Bates and Maechler, 2009; Bates et al., 2014). The LMM's provided a non-parametric approach to explain variability in the response variables by fixed effects (factors that were included in the study design) and random effects. Random effects account for factors which were not part of the study design, but possibly affected variability in DOC, pH and EC. The fixed effects considered in this study were groundwater condition (GW - DRIP, non-DRIP), position

in the landscape relative to the stream (POS – riparian, transition, upland), season when the samples were taken (TIME – spring, summer, autumn), and the two-way interaction between GW and POS and TIME respectively. The included random effects were the stream identity and the transect identity along which the wells were situated. In this way we accounted for specific catchment and hillslope properties. The model structure selection was based on the lowest AIC (Akaike's Information Criterion).

We evaluated the model performance using Type II Wald F tests with Kenward-Roger degrees of freedom (since all explanatory variables are factors). F statistics indicate the explained variance as a ratio of unexplained variance. An effect was considered significant if p-values <0.05. We evaluated the assumption of Gaussian distribution of errors by inspecting residuals and quantile distributions. For DOC five outliers, and for pH two outliers were removed from the upper quantile. For EC one in the lowest tail and two in the highest tail of the distribution. For comparing contrasts of levels within explanatory factors

(for example DRIP vs. non-DRIP comparisons), we investigated least square means using R-package *lsmeans*, including Tukey adjustment to account for potential differences in sample size (Lenth and others, 2016). Furthermore, the marginal and conditional coefficient of determination ($R^2_{mar}$ and $R^2_{con}$) was presented to compare explained variance by the fixed effects, and the variance explained by the fixed and random effects together (R-package *MuMln*)(Barton, 2014).



## 3 Results

### 3.1 DOC

The water collected in wells situated in the DRIPs had a higher mean DOC concentration (33.9 mg/l) compared to non-DRIP wells (19.9 mg/l, Fig. 3). The position gradient had higher explanatory power (Table 1), but differences were much smaller

compared to the differences between DRIPs and non-DRIPs (from 22.8 mg/l to 28.2 mg/l, DF=327, p=0.0001). In the upland wells there was no statistical difference in DOC concentrations between DRIP and non-DRIPs (p=0.1844, Fig. 4 upper left panel), even though mean DOC concentrations are contrasting (29.2 mg/l and 16.4 mg/l for DRIP and non-DRIPs, respectively). DOC concentrations in DRIPs increased towards the riparian wells (36.3 mg/l), while in non-DRIP riparian wells DOC concentration were much lower (20.1 mg/l, DF=19, p=0.03), gaining only 3.7 mg/l from the upland to riparian

wells. Therefore the overall gain in riparian DOC concentrations was most accountable to DRIPs (29.2 mg/l to 36.3 mg/l, DF=326, p=0.0003). In terms of seasonality, there was a weak yet significant effect on DOC concentrations when considering all the wells together (Table 1, TIME, p=0.054). However, just as the position gradient, there was also an interaction between GW and TIME explaining variability in DOC concentrations. In summer and autumn DRIPs had double as high DOC concentrations (36.4 and 33.3 mg/l), compared to non-DRIP areas (18.0 and 17.7 mg/l, Fig. 5, upper panels). However, during

snowmelt in spring, this difference disappeared. This change was a result of an average 20% decrease in DOC concentrations in DRIPs (28.5 mg/l) compared to the summer average. In non-DRIP areas there were no such significant contrasts, although there was a small, increase in spring (21.6 mg/l) compared to summer and autumn (18.0 and 17.7 mg/l, p=0.4986 and p=0.3019). Overall, the fixed effects alone explained 22% of the variance in DOC found in the groundwater well network. With the random effects included the explained variance was 68%.

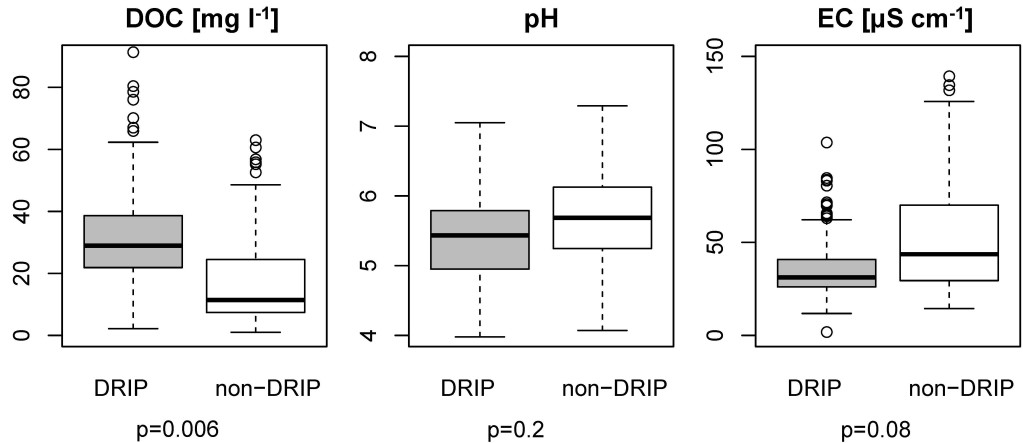


**Figure 3 Groundwater chemistry of DRIP versus non-DRIP. Each boxplot represents one response variable. The water chemistry of the entire well network is separated based on the riparian hydrological condition. DRIPs are wet riparian areas with a large contributing area (left-hand side, in grey). Non-DRIPs are drier riparian areas with mostly local hillslope contributions (right-hand side).**

### 3.2 pH

Although typically associated with DOC, the pH was not as distinctly different between DRIP and non-DRIP water as DOC (Fig. 3). Overall the fixed effects accounted for 13% of the variance, and 55% including random effects (Table 1). Mean pH levels were 5.38 for DRIPs and 5.66 for non-DRIPs (DF=16, p=0.2). Instead, position in the landscape had more effect on the variability in pH: the upland pH was similar at DRIPs and non-DRIPs and decreased towards the riparian area (5.66 to 5.40,

P<0.0001). Although no significant effect was found for interaction between the landscape position and hydrological conditions (Table 1), the least square means analysis showed a pronounced decrease in pH from upland to riparian wells in the





DRIP areas (5.57 to 5.19, P<0.0001, Fig. 4 middle panels). The second important explanatory variable was seasonality (TIME in Table 1). The most notable was the increasing pH from the summer to autumn (5.37 to 5.70, P<0.0001), both in DRIP and non-DRIP areas (Fig. 5, center panel and center-right panel). In the transition to spring, pH decreased again ($pH_{spring}$=5.48), mostly due to a shift in the DRIPs (p=0.04). Furthermore the variability in pH in non-DRIP water was high compared to DRIP

areas, especially during summer (Fig. 5, center plot).

**3.3 EC**

Mean electrical conductivity from DRIP water was 36.2 µS/cm, which was lower (p=0.08) compared to the mean of non-DRIP water (51.6 µS/cm, Fig. 3). The variance in EC was mostly explained by POS and TIME, and the interaction between GW and POS (Table 1). Overall the conductivity increased from the upland to the riparian wells (39.3 to 48.0 µS/cm) and increased as

well from spring to autumn (39.7 to 48.7 µS/cm, Fig. 4 lower panels). The interactions between groundwater conditions and the position relative to the stream were mostly related to two specific contrasts. The variability in EC in non-DRIP groundwater increased from the upland to riparian wells, while in DRIP areas the EC remained stable (Fig. 4, bottom row). Moreover large differences were found between DRIP and non-DRIP in the riparian wells, where the EC in non-DRIP riparian areas was twice as high as the EC in DRIPs (63.6 µS/cm compared to 32.4 µS/cm). In the upland areas, the DRIP and non-DRIP water was

similar. Non-DRIP water increased from 40.5 µS/cm to 62.4 µS/cm from the upland wells towards the riparian wells, while DRIPs even decreased in conductivity (38.2 and 32.4 µS/cm for upland and riparian wells). Over the different seasons, the contrasts between DRIP and non-DRIP chemistry were consistent (Fig. 5, lower panels). The interaction between groundwater and seasonality was not found to have an effect on EC. The only specific contrasts for both DRIP and non-DRIP was a 5µS/cm decrease from autumn to spring ($P_{DRIP}$=0.05, $P_{non-DRIP}$=0.0007). Overall, the explained variance of our LMM was 70% for EC,

compared to 22% when only accounted for fixed effects (Table 1).

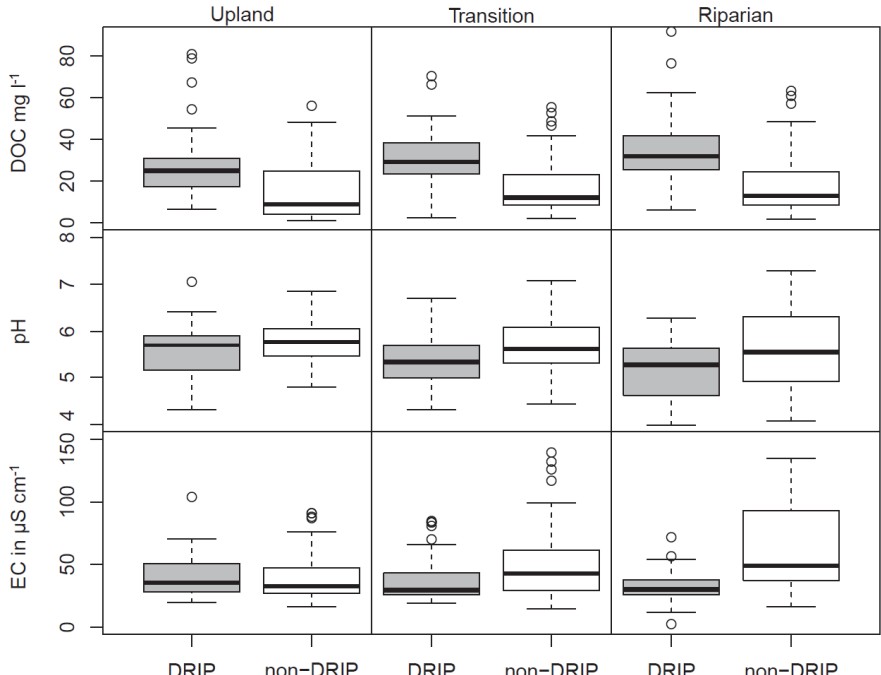

**Figure 4 Groundwater chemistry gradients from upland to riparian wells. In each column DOC, pH and EC are presented for a location relative to the stream (Riparian, Transition and Upland). Within each panel DRIP boxplots are presented in grey and non-DRIP boxplots in white.**





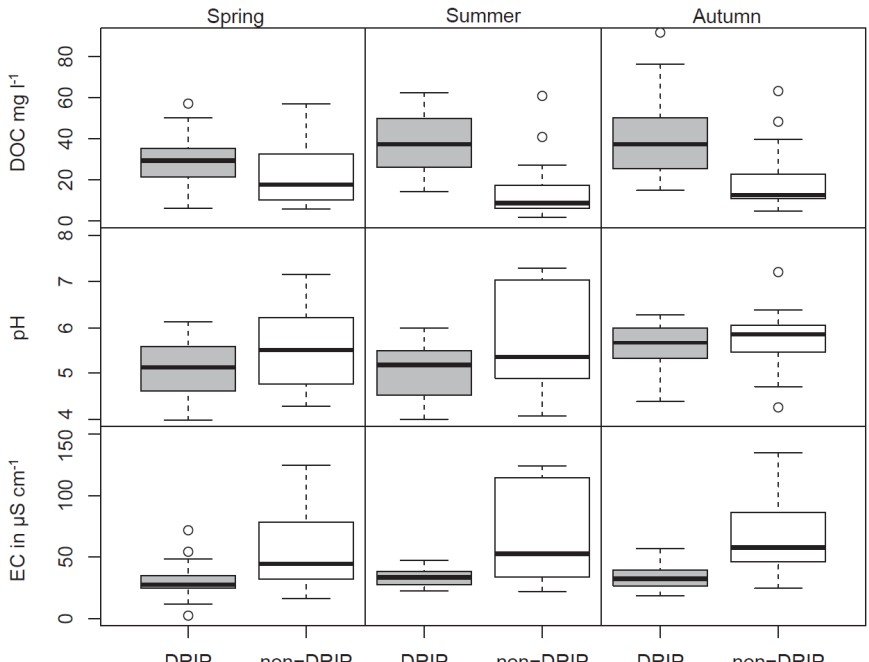

**Figure 5 Groundwater chemistry in a seasonal gradient of the riparian wells. In each column DOC, pH and EC are presented for the spring, summer and autumn season. Within each panel DRIP boxplots are presented in grey and non-DRIP boxplots in white.**

## 4 Discussion

Our results showed that riparian groundwater is highly variable in its chemical composition throughout space and time. However, we found that parts of this variability cannot only be assigned to commonly used factors, the distance to the stream and seasonality, but also to hydrological pathways (DRIP vs. Non-DRIP). DOC concentrations in DRIPs were twice as high compared to the less hydrologically active riparian areas (non-DRIPs). The groundwater chemistry of non-DRIPs was characterized by high electrical conductivity, and increasing variability towards the stream and higher temporal variability.

Differences in pH were less distinct, and mostly accountable to seasonal changes. These results confirm our hypothesis that DRIPs have a more organic groundwater chemistry, while non-DRIP water can be associated with more mineral chemistry. Furthermore, DRIPs had a more constant and stable biogeochemical character compared to non-DRIP riparian area across space and time.

The link between groundwater chemistry, mobilization of (old) groundwater and stream chemistry has puzzled hydrologists, especially when it comes to chemical variability during hydrological events (Kirchner, 2003). The transmissivity feedback mechanism of till soils has been pointed out as a possible resolution of the double paradox for boreal headwaters in the Krycklan catchment (Bishop et al., 2004; Laudon et al., 2004a). On hillslope scale, the dominant source layer (DSL), a highly conductive layer just under the soil surface, plays an important role as rising groundwater rapidly mobilizes 'old water' in the

unsaturated zone, and spatially connects various sources of soil water (Ledesma et al., 2015). Across the RZ of headwaters, earlier work has already pointed out that spatial groundwater variability is linked to groundwater conditions and the composition of riparian soils (Grabs et al., 2012). It has also been demonstrated that wet riparian areas similar to DRIPs dictate stream DOC dynamics (Creed et al., 2003; Werner et al., 2019). The contrasting chemistry of DRIPs and non-DRIPS presented here further supports the link between hydrological pathways and variability of groundwater chemistry, and possibly explains

why pre-event water is so quickly mobilized. On event basis, the stream chemistry that is monitored at a single downstream



point is an integration of different hydrological responses of DRIPs (Leach et al., 2017; Ploum et al., 2018), and the activation of non-DRIP hillslopes. This possibly explains why we encounter chemical variability of old water in the stream, as different pre-event water from DRIP and non-DRIPs discharges into the stream. Interestingly, we found that during spring flood conditions, the high DOC in DRIP groundwater decreased and became less spatially variable. In contrast, non-DRIP water

increased in DOC and in variability during this season. In the case of the DRIP groundwater, we believe that snowmelt dilution is a likely cause for the decreased DOC concentrations during spring. Furthermore, ice sheet formation in the DRIP areas has been reported previously, which can route water over the ice surface instead of the organic rich subsurface flow paths, such as the DSL (Ploum et al., 2018). These overland-flow findings are similar to dilution effects and soil frost effects reported for wetland dominated streams during spring floods (Laudon et al., 2004a, 2011). The non-DRIP areas are typically drier, only

drain local hillslopes, and have lower groundwater levels during drier periods (summer). During the spring, groundwater levels rise and water flows through shallow organic soil layers. If DRIP and non-DRIP riparian zones become more similar, why is there no stabilizing and decreasing pattern in DOC in forested streams during spring flood (Laudon et al., 2011)? Given that water transported during high flow events is predominantly old, pre-event soil water, an increased EC could have been expected as the old water has relatively longer contact time with the soil prior to mobilization. However, we found no increase in EC

during spring, which could lead back to the second part of the double paradox: most of the time non-DRIPs have low water tables, but in spring, the organic rich DSL is spatially connected, mobilizing old water that has unlikely been in contact with the mineral subsurface. Although our findings might not be able to provide any resolutions, showing the differences between DRIP and non-DRIP here suggest that there is a further distinction in hydrological functioning within boreal forests, similar to the governing hydrological theories differentiating between forest- and wetland dominated catchments (Laudon et al., 2011).

The further analysis of event and pre-event water of DRIP/ non-DRIP contributions could shed light on open questions around stream chemistry and runoff generation in boreal headwaters.

The difference in chemistry observed between DRIP and non-DRIP areas within these headwater streams also promotes a new conceptual model of boreal system hydrology and biogeochemistry on catchment level (Laudon and Sponseller, 2018). A

preliminary analysis showed that 57% of the Krycklan catchment is draining into the stream network through DRIPs, spatially covering only 12% of the riparian zone (W. Lidberg, personal communication). Previous work has demonstrated that in boreal catchments the input of deeper/older groundwater (with high EC) increases with drainage area, up to a threshold where old and new groundwater input reach a balance (Peralta-Tapia et al., 2015). The presence of DRIPs in headwaters can play an important role in the balance of old and young water further downstream. Since DRIPs provide low EC water, which

presumably is young, and non-DRIPs convey older water, stream reaches with shortage or lack of DRIPs would introduce a larger proportion of older water. On the contrary, DRIP dominated headwater catchments likely are dominated by young groundwater. The chemical contrasts of DRIP and non-DRIPs potentially changes the development of chemical patterns across scales observed by Peralta-Tapia et al (2015). Moreover, the presence of DRIPs possibly disrupts the mixing process of old and young water in stream networks, making it a longer process to reach chemical stability. Of course, the presence of DRIPs

(and their initiation threshold) is tightly coupled with geological setting, as predominantly post glacial till areas promote shallow, horizontally dominated groundwater flow. In addition, their control on streams of higher order is likely coupled to the degree of chemical contrast and their hydrological activity. Previously, links between hydrological pathways on groundwater chemistry dynamics have been found to significantly affect the chemistry of a fifth order river (Carlyle and Hill, 2001). Further, flow paths known as watertracks have been shown as important biogeochemical controls on higher order Arctic

rivers (Harms and Ludwig, 2016; McNamara et al., 1999).

Along the stream networks, the delineation of DRIP/non-DRIP areas in the riparian zone can help to implement hydrologically adapted buffers in forest management, ensuring that waterbodies remain their water quality (Kuglerová et al., 2014b; Tiwari





et al., 2016; Wallace et al., 2018). Traditionally, forest practices considered fixed width buffers even though the riparian function is not homogeneous around all water bodies (Buttle, 2002). For example the extent of riparian soils vary, which showed to be of major influence on terrestrial carbon exports to streams (Ledesma et al., 2018). Also species richness within the RZ is reliant on many local factors (Kuglerová et al., 2014a). Furthermore, the temporal expansion and shrinkage of stream

networks and the subsequent contributing riparian area are generally not homogenous (Ågren et al., 2015). Our results support that also in terms of groundwater chemistry, variable widths should be considered in buffer management. We found that DRIP water had already a distinct chemical signature before entering the RZ: 80% of the DOC originated from upland riparian wells. This suggests that the chemical role that is associated with RZ's, extends further away from the stream than the traditional fixed-width buffer management considers. Recent advances based on machine learning offer novel tools to implement

hydrologically adapted buffer management on regional level (Lidberg et al., 2019).

For identification of control points, improving hydrological models and sustainable forest management practices, a binary approach with little need of local properties can be a very useful tool. However, to understand the underlying mechanisms and the link to the landscape, hydrology of RZ's should be considered non-binary (Klaus and Jackson, 2018). Our LMM's showed

that a large part of the variance is explained by the random effects, which contain information regarding the unique properties of individual transects and to a lesser extent the subcatchments. The large variation in non-DRIPs lead to statistically weak contrasts, but this does not mean non-DRIP RZ's are less important. It demonstrated that an important chemical change also occurs in riparian non-DRIP areas, of which the complexity overpassed the binary simplifications we have made in this study design. There is not a clearly defined threshold when a wet riparian area can be considered as a control point for stream

dynamics. Not only in terms of hydrological flow paths, but also in the wetness state (which likely dictates a large part of the chemical characteristics). Moving from a topography-driven binary approach towards process-based analysis can contribute to understanding the mechanisms behind the contrasting biogeochemical characteristics of the RZ. Here we have shown this for two extremes in terms of hydrological pathways. Next steps can, for example, include local landscape characteristics, subsurface soil properties and groundwater level dynamics to further decipher whether soil, biology or hydrology define the

biochemical characteristic throughout the RZ. This means that local subsurface conditions across soil horizons and landscape features such as slope, land cover and/or aspect might be able to explain a significant part of the processes that generated spatiotemporal variability in groundwater.

## 5 Conclusions

Are hydrological pathways and variability in groundwater chemistry linked in the riparian boreal forest? Yes, based on our

findings there is a strong link between the hydrological connection of the riparian zone, and the groundwater chemistry sampled in different seasons. Hydrological pathways confluence in the riparian zone at Discrete Riparian Inflow Points (DRIPs), where we found organic-rich, stable groundwater chemistry compared to the remaining, drier riparian zone. Combining their chemical characteristic and hydrological importance for headwaters, we propose that DRIPs are *control points* in the boreal riparian forest. To our knowledge, this study is the first to characterize spatial groundwater chemistry that a priori incorporated the

hydrological pathways in the riparian zone in the study design.



However, to fully evaluate their impact on stream water generation and the associated stream chemistry, there is the need to further investigate the hydrological activation, and a broader chemical characterization. To understand the mechanisms and processes that link hydrological pathways and groundwater chemistry in boreal forest, we suggest to move towards non-binary approaches incorporating groundwater fluctuations, soil properties and landscape characteristics.

## 5 Data availability

Data is available upon reasonable request through the first author (stefan.ploum@slu.se). Krycklan data is openly available through the Svartberget database: https://franklin.vfp.slu.se/

## Author contributions

LK and HL conceptualized the study design and methodology, supervised data analysis, interpretation and writing process. AP conducted the well installation, provided field support, reviewed the written text and was involved in discussions. SP was responsible for collection of data and data analysis, figures, interpretation and writing.

## Competing interests

HL is a member of the editorial board of this special issue of Hydrology and Earth System Sciences.

## Acknowledgements

This study was funded by Oscar and Lili Lamm Foundation and Svenska Forskningsrådet Formas, but the Krycklan catchment is also supported by SITES (VR), SKB, KAW and the Kempe Foundation. We acknowledge Anna Lupon and the staff at Svartberget research station for field support. We thank Jason Leach for contributing to the discussions and hydrological interpretations.

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





**Tables**

Table 1 Summary of statistics from LMM models for DOC, pH and EC. The three columns show the response variables DOC, pH, and EC. The upper two rows show the marginal and conditional coefficient of determination ($R^2_{mar}$ and $R^2_{con}$), which explain variance by the fixed effects, and the variance by the fixed and random effects together. For each explanatory variable and the interaction with GW, the p-value and F-statistic is presented. GW differentiates between DRIP and non-DRIPs. POS represents the three positions in along transects being: riparian, transition and upland. TIME represents the three different seasons when sampling has taken place: spring, summer and autumn. Significant codes: $p < 0.001$ '***', $0.001<p<0.01$ '**', $0.01<p<0.05$ '*', $0.05<p<0.1$ '.', $p>0.1$ '-'. Explanatory variables with a 'variable1:variable2' represent the interaction between both variables.

| | | DOC | pH | EC |
|---|---|---|---|---|
| $R^2_{mar}$ | | 0.22 | 0.13 | 0.21 |
| $R^2_{con}$ | | 0.68 | 0.55 | 0.70 |
| | | | | |
| GW | p-value | 0.012 (*) | 0.20 (-) | 0.052 (.) |
| | F | 8.47 | 1.99 | 4.36 |
| POS | p-value | <0.0001 (***) | 0.0001 (***) | <0.001 (***) |
| | F | 10.02 | 9.24 | 7.08 |
| TIME | p-value | 0.054 (.) | <0.0001 (***) | <0.0001 (***) |
| | F | 2.95 | 13.48 | 11.31 |
| GW:POS | p-value | 0.18 (-) | 0.11 (-) | <0.001 (***) |
| | F | 1.70 | 2.24 | 32.11 |
| GW:TIME | p-value | <0.0001 (***) | 0.75 (-) | 0.49 (-) |
| | F | 12.07 | 0.288 | 0.72 |