# Peer review of "Are DOC concentrations in riparian groundwater linked to hydrological pathways in the boreal forest?"

_Hydrology and Earth System Sciences, 2019_

## Referee Comment (RC1) · Anonymous Referee #1 · 16 Aug 2019

General comment This is a concise paper that compares groundwater chemistry in sub-catchments of the Kryckland observatory and asks questions about chemical variability in relation to hydrological pathway. The paper is written in a clear and concise way with not much to complain about in the introduction and results. The paper clearly shows and describes the variability of groundwater chemistry in relation to its hydrological activity. This is a very relevant result that is worth being published in HESS. However, I have some point of criticism that should be addressed: Groundwater was sampled with fully screened wells which means that there is an unknown integration of water quality over depth potentially changing between sampling campaigns. The authors should carefully discuss the pros and cons of this approach. Later on in the

text there seems to be a not-outspoken assumption of vertical homogeneity of ground-water quality and a high weight on lateral heterogeneity. This needs to be made more clear and concise as this new DRIP-concept seems to stand against the former RIM-(Seibert et al. 2009) and DSL-concepts (Ledesma et al. 2015) that were derived from the same study site. The authors used linear mixed-effect models (LMM) to analyze spatiotemporal patterns of groundwater chemistry. There should be some effort in the text to justify the choice of this method and also on the choice of predictors. One could argue that the authors did not look hard enough to include other predictors (e.g. local TWI) but instead chose a method which can handle random effects from unaccounted factors. Finally, I have a major concern with the discussion section that is rather an im-plication section centered on the question what the varying groundwater quality would mean for stream chemistry. However, this was not part of the study design. So the dis-cussion should rather be focused around the question, why the measured groundwater quality was different in different parts of the riparian zone.

What came into my mind here: Aren't DRIPs just an extension of the fractal stream network into the catchment. DRIPs are along topographic depressions funneling water flow in the same way as the stream network. The major difference is that they are not (permanent) pathways of surface flow but rather pathways of shallow subsurface flow. Maybe I missed that connection in earlier studies that were cited here. I however very much like the idea to refine the view on riparian zone by that type of concepts focused around major flowpaths. In essence I support this manuscript but ask for a substantial revision of the discussion section and a better justification of field and statistical methods.

Specific comments

Abstract:

L13: This sentence is potentially misleading. What is meant by chemical variability in the riparian boreal forest? The linkage of riparian groundwater as described before

and forest is not clear.

L20: The pairing of hydrological connection and groundwater condition cannot be understood here. Can you find a more telling factor name?

L20: The water provided by DRIPS – is that surface water discharging from the DRIP or groundwater within the DRIP?

L23: "chemically more stable" may be misleading. Do you mean spatially and/or temporally homogeneous?

Introduction

P2L1: While I in general like such comparisons the idea of headwaters as "capillaries" was not immediately clear to me.

P3L10-13: These two sentences need some references.

Methods

Fig. 2: The catchment delineation is somewhat distracting. I suggest to limit the catchment boundaries to the catchment with studied DRIPs/ nonDRIPs. What determines the catchment outlet? A gauging station? Maybe show them to make clear that this was not completely arbitrarily chosen.

P5L6: TWI units are not [ha] – what do you mean by "topographic wetness index (2 ha threshold)"?

P6L1: Can you specify the wells in terms of diameter and material? What is the meaning of a groundwater sample taken in a fully screened well? Do you expect this to be a representative sample from all depths or rather a sample from the most conductive depth? In the latter case the depths where most water is coming from may change over time with changing groundwater levels. I would like to see a critical evaluation of your sampling design and drawbacks (assets?) of the chosen methods! The methods chapter is likely not the right place.

Results

P7L4: I don't understand why you jump in directly with that statement in the second sentence already.

P7L26: Did you expect more distinct differences? Keep in mind that the pH is a logarithm of concentrations and small changes can mean a lot compared to DOC and EC.

P8L16: When you talk about seasonality you mean factor TIME, right? It would be helpful to stick to those factor names and provide it in brackets if other words such as seasonality are used.

Fig. 4: I suggest that you indicate if the mean is significantly differing in individual panels. That would improve readability here. Same is true for Fig. 3 and 5

Discussion

The discussion male strong links of groundwater spatial variability to stream chemistry temporal variability. But this is not what was shown in the results. I miss a discussion of why groundwater quality was as it was measured and presented above. All the discussion is rather focusing on implications.

P9L23 to P10L3: I have problems following the argumentation here. Why does the contrasting chemistry of DRIPS and non-DRIPS explains why pre-event water is quickly mobilizes? Do we really need DRIPS and non-DRIPS to explain temporal variability of stream chemistry within an event? That is also covered by vertical chemical heterogeneity (taking Seibert's RIM for instance).

P10L26: The generalization (that totally make sense) of your findings to the larger scale of the Kryckland catchment is a selling point. I suggest to base the statement made here on DRIPS in catchments on a sound and reproducible analysis and not on a personal communication.

---

## Referee Comment (RC2) · Anonymous Referee #2 · 7 Sep 2019

I have reviewed the draft manuscript: 'Are hydrological pathways and variability in groundwater chemistry linked in the riparian boreal forest?' submitted by Ploum et al for possible publication in HESS. I like the general premise of this study, eg that preferential flowpaths from hillslopes through riparian zones need to be better considered when characterizing the baseflow controls on stream chemistry and dissolved organic carbon availability (DOC). Too many riparian studies are based around 'uniformed' or random piezometer transect designs, and without the hydrological flow context, the groundwater chemistry data are difficult to interpret. I do think this material is appropriate for HESS, though the paper could benefit from a change in emphasis and additional heat tracing data that it seems the authors may have already collected.

[Figure]

Currently, the primary question addressed by the study is posed as: 'Are hydrological pathways and variability in groundwater chemistry linked in the riparian boreal forest?' There have been many studies to document strong variance in groundwater chemistry as controlled by varied advective flowpaths, too numerous to list here. The current study by Ploum and all is unique from many of these as the preferential flowpaths in question source varied dissolved constituents to headwater streams. I suggest the authors refocus the main question to something like: 'Do DRIP's represent preferential conduits of DOC-rich groundwater to headwater streams in a boreal forest?' Or something at least more specific to this study than 'variability in groundwater chemistry'. It seems that the most compelling data presented in this study show the 'DRIPs' in this boreal headwater system are enriched in DOC, which presumably results in higher flux of DOC to the channel via preferential shallow groundwater discharge compared to more diffuse flow through till (though hydrological fluxes are not actually measured or inferred here). Further, the authors document interesting temporal trends in DOC and SpC, the latter being much less meaningful without additional chemical characterization.

In general, I feel the LMM statistical analysis was appropriate for assessing DRIP/non DRIP well DOC, pH, and EC. The results between these binary classifications are interesting, though for all the effort on well installation and sample collection some basic hydrological data seem missing. Where lateral gradients measured between wells and the stream? Were any hydraulic conductivity tests performed? Do we have any idea of groundwater flow rate/flux to the stream from DRIP vs non-DRIP zones? This is a flowpath-based study but the reader is left without any real flow-based hydrogeological information. The addition of some basic quantitative hydrogeological data, and perhaps some additional measured parameters such as dissolved oxygen, could have nicely increased the impact of this study. Without any evaluation of piezometer water age (eg dissolved gas-, isotope-based) the discussion of old vs new water contributions is highly speculative and should be scaled back. I really like the concept you put forth of DRIPs as drivers of younger water fractions in streams where low permeability soils

dominate (eg tills), though it is difficult to support this using EC as the primary parameter. Despite these criticisms I do think this work could make an important addition to HESS after some revision considering the major comments below and that of the other reviewers.

The statement is made toward the end of the Introduction: 'However, in order to determine whether DRIPs matter for stream biogeochemistry, chemical characterization of the discharging groundwater is needed.' Yes, but this is only half of the equation, the other being the flux of groundwater to the stream which was not evaluated here in any way. It seems at least some data specific to groundwater discharge was collected in these streams and presented by Ploum et al 2018. Where many of the DRIPs instrumented here with piezometers identified in the stream as preferential discharge points using heat tracing? If so that data could be briefly included here with an additional figure, and go a long way to convincing the reader that the wet topographic low points mapped here as DRIPs are actual preferential flowpaths from the hillslope to the stream. Without any such thermal or hydrological gradient/permeability data it is difficult to accept with confidence that the DRIPs mapped here are actual preferential flow zones, compared to the surrounding soils. To be clear I think that the hydrogeologic interpretation of DRIPs made by the authors is likely generally correct, particularly after reading/watching Ploum et al 2018, but the current paper would benefit greatly from some groundwater flow-based data. I list several more major comments below (I realize some are a bit redundant with this narrative) followed by some more minor text suggestions:

1. A main conclusion of this work is stated as: 'We concluded that hydrological pathways and spatial variability in groundwater are linked, and that DRIPs are important control points in the boreal landscape.' Can you build on this statement in the abstract to be more specific to your study? Near stream shallow chemical variability has long been linked to flowpaths. Perhaps comment more specifically in the abstract regarding the spatial variability you observed to set your work apart from previous studies. The

Results section discusses some temporal patterns, but I do not see this data reflected in any of the figures or tables. You could develop a figure specific to the interesting temporal patterns, this information is shown somewhat in Table 1 'TIME' variable analysis but could be shown more explicitly. Also, I think the fact that your research indicates DRIPs our important DOC pathways to the boreal stream corridor is quite important.

2. Under your definition, are DRIPs only driven by surface topography and wetness? There are numerous instances of preferential discharge of groundwater and interflow through the riparian zone through highly permeable sediments that are not correlated with surface topography, and in glacial sediments often occur at local topographic high points (sand and gravel deposits transecting the riparian zone). I agree that local topographic depressions often lead to saturated conditions in the riparian zone, but that is not the same thing as strong hydrologic connectivity to the stream channel, which depends on the combination of lateral gradient and sediment permeability. Previous work by this group (Ploum et al 2018) used heat tracing methodology to locate/confirm preferential discharge of riparian water to the stream channel, which makes sense. However the current work does not seem to tie the definition of DRIP to actual observed high-discharge points, which I think is unfortunate. Not all saturated depressions will be points of preferential discharge to the channel, which strongly depends on soil permeability. Further, according to your statement: 'water in DRIPs travels a longer distance horizontally; in presumably wet, highly permeable, organic rich soil." It seems your definition of DRIP is relatively narrow, and based around forested headwaters similar to where your study has been conducted. I think it would be quite helpful for you to more specifically define 'DRIP' early in the manuscript (in the Introduction), and acknowledge that this definition applies to only a subset of preferential riparian groundwater discharge zones in headwater systems. Your broad definition of DRIP in section 2.2 (eg 'groundwater discharge zone, groundwater hotspots, cryptic wetlands, swales, focused seepage, discrete seepage, springs, upwelling zones, preferential discharge, ephemeral streams and zero-order streams') does not seem to apply to the functional definition you apply here for shallow lateral flow above the mineral soil horizon, so

please be clear on what definition you are using for this study.

3. It is not clear to me how DRIPs are defined for 'upland' areas... do you use the same definition based on topographic depression and wetness index that you use for the riparian zone DRIP areas?

4. Abstract L10 and elsewhere: it is somewhat of a misnomer that all riparian flowpaths lead to biogeochemical alteration of discharging water chemistry. Low-carbon mineral soils and highly preferential flowpaths such as peat pipes and other macropores can yield little alteration in hillslope and deeper groundwater discharge. In fact your findings indicate that DRIPs lead to less net reaction then more diffuse groundwater discharge through the riparian zone: 'Moreover, DRIPs were chemically more stable from the upland area to the stream'. You might check out this commentary for some relevant discussion : (https://onlinelibrary.wiley.com/doi/10.1002/hyp.11153)

5. It would seem fully screened wells down to apprx 1.5 m depth would integrate shallow 'DRIP' water and deeper mineral soil water, how did you account for this? I think your statements in the Discussion section regarding 'old' and 'young' water are a bit to speculative, as this is essentially only based on EC data, a parameter influenced by a number of flowpath process. It does not seem like any age dating/isotope analysis was performed, so how confident can you be regarding relative water ages/residence times? Also, you mention the piezometers were installed until they reached a hard layer. Did this depth vary systematically from DRIP to non-DRIP locations? If so you could include that data, as depth to rock/confining layer can also be a strong control on shallow groundwater flowpath chemistry.

Fig 1: Although we might expect local shallow percolation in non-DRIP near stream zones, groundwater flow is likely dominated by the lateral component toward the stream (in gaining stream reaches), though the discharge magnitude may be reduced compared to preferential discharge points. I suggest you alter the 'vertical groundwater flow' language in the 3rd panel of this Figure, the vertical flow you refer to may instead

by non saturated percolation toward the water table, where groundwater flow is predominantly horizontal. Have you measured any vertical head gradients at the wells, and lateral gradients between wells, to support these conceptual diagrams? Fig 3- The caption could be simplified, you do not need to define DRIPs in the caption as this is done in the text

Minor points: Pg 2: L2 repeat of the word 'landscape', please look for replacement L3 I am not sure what you mean by 'newly introduced water', can you be more specific? L19 do you have a reference example to associate with: 'Traditionally, streamflow generation has often been assumed to be driven by spatially diffuse groundwater exchange often released at a constant rate.'? pg 4 L20: could cite here the hydrographs shown in Ploum et al 2018 L24: you would not consider the fall period to be 'baseflow' dominated as well or is this just a winter condition in your watershed system? Pg 7 L13: replace 'double as high'

---

## Short Comment (SC1) · 19 Sep 2019

Variability in landscapes is a challenge for understanding how landscapes influence water chemistry in space and time. By developing a sampling design based on a hypothesis about how water is flowing through the riparian zone, this study has provided new insights into the hydrobiogeochemical structures that shape the connection between landscapes and waters. This has practical implications for the design of buffer strips that are widely used in water management. As such this paper can be a valuable contribution to the literature. I think its value would be enhanced if a few points in the paper got further attention.

[Figure]

One concerns the distinction drawn between DRIPs and the confluences of ephemeral streams on page 5, lines 19-21. The text here was not clear. I think the authors are trying to say that such confluences are not included in their definition of DRIPS since DRIPS do not have clear channels. It would be good if this could be clarified.

A second point I suggest that the authors address concerns the discussion of relative contributions from DRIPs and Non-DRIPS to stream chemistry under different flow conditions (page 10, lines 29-34). This part of the discussion talks about the contrasting chemistries coming from DRIPS and non-DRIPS. But the effect of chemical differences in the source waters on stream chemistry depends on the proportion of water coming from the different source waters. Is there some assumption underlying this part of the discussion about how much water comes from DRIPs relative to non-DRIPS during high and low flow conditions? Clarification of that would help make the points in this part of the discussion more persuasive.

Furthermore, if the DRIPS do not include the confluences of intermittent streams with the perennial stream channel, it would be important to mention what these ephemeral streams are doing to contribute to the high-flow stream chemistry being talked about in the discussion.

One final question, the concept of a "DRIP initiation threshold" is mentioned on page 10 line 35, but a definition of what this means is not given. Please explain the term. .

---

## Author Comment (AC1) · 16 Oct 2019

General comment

This is a concise paper that compares groundwater chemistry in sub-catchments of the Kryckland observatory and asks questions about chemical variability in relation to hydrological pathway. The paper is written in a clear and concise way with not much to complain about in the introduction and results. The paper clearly shows and describes the variability of groundwater chemistry in relation to its hydro- logical activity. This is a very relevant result that is worth being published in HESS.

[Figure]

Answer: Thank you for evaluating the manuscript and the constructive commentary. Below we will present answers to each point that is addressed.

However, I have some point of criticism that should be addressed: Groundwater was sampled with fully screened wells which means that there is an unknown integration of water quality over depth potentially changing between sampling campaigns. The authors should carefully discuss the pros and cons of this approach.

Answer: We agree with Referee #1 that we can further elaborate on the well infrastructure, sampling protocol and the subsequent effects on the water chemistry of each sample. We will insert on page 2 in line 2 the following: 'We assumed that the water sampled from the well is a weighted average of the phreatic aquifer, down to the depth of the well. Given the exponentially decaying hydraulic conductivity with depth, this assumption would imply that, under saturated conditions, the majority of the water is therefore from the upper soil layers, referred to as the dominant source layer (Ledesma et al., 2015). We assume that the lateral flow below the well bottom is negligible compared to the flow in the shallow subsurface.'

Later on in the text there seems to be a not-outspoken assumption of vertical homogeneity of ground- water quality and a high weight on lateral heterogeneity. This needs to be made more clear and concise as this new DRIP-concept seems to stand against the former RIM- (Seibert et al. 2009) and DSL-concepts (Ledesma et al. 2015) that were derived from the same study site.

Answer: We agree that further elaboration is needed, in combination with the previous comment, regarding the vertical heterogeneity in the groundwater quality and the associated RIM and DSL concepts. Our study could be considered complimentary to the existing literature, rather than opposing. To clarify this we will add on page 2, lines 21-24: 'In terms of conceptualizing the spatial heterogeneity of groundwater inputs to streams (both hydrologically and biogeochemically), the RIM model and DSL concept have considered the vertical heterogeneity in riparian groundwater fluxes to

boreal streams (Ledesma et al., 2015; Seibert et al., 2009).'

The authors used linear mixed-effect models (LMM) to analyze spatiotemporal patterns of groundwater chemistry. There should be some effort in the text to justify the choice of this method and also on the choice of predictors. One could argue that the authors did not look hard enough to include other predictors (e.g. local TWI) but instead chose a method which can handle random effects from unaccounted factors.

Answer: The use of LMM was justified by the setup of the groundwater well network. Given that this is, to our knowledge, the first groundwater chemistry database that has been designed and compiled a priori based on three factors (hydrological condition, distance to stream and season), we deem those factors as our only true testable factors. Since the study sites were selected partly based on TWI, TWI and DRIP vs. non-DRIP are inherently correlated. Given the large improvement of the model by the inclusion of random factors which are describing spatial dependency of the samples, we discussed that there is a large part of spatial variability in groundwater chemistry left to be explained by factors that are not included by our study. We deliberately attempted to explain groundwater chemistry in a simple manner with generic and a priory set predictors, and explaining variability by unique properties of the study sites would impair any upscaling of our findings.

Finally, I have a major concern with the discussion section that is rather an implication section centered on the question what the varying groundwater quality would mean for stream chemistry. However, this was not part of the study design. So the discussion should rather be focused around the question, why the measured groundwater quality was different in different parts of the riparian zone.

Answer: We agree that the discussion is rather speculative and conceptual. We will balance the interpretation of our findings more appropriately with the potential implications. Especially, we will focus more on explaining and discussing the variation in groundwater quality in our well network while keeping the implications to stream chemistry to lesser extent. We will elaborate on the larger context of our work in a short conclusion.

What came into my mind here: Aren't DRIPs just an extension of the fractal stream network into the catchment. DRIPs are along topographic depressions funneling water flow in the same way as the stream network. The major difference is that they are not (permanent) pathways of surface flow but rather pathways of shallow subsurface flow. Maybe I missed that connection in earlier studies that were cited here. I however very much like the idea to refine the view on riparian zone by that type of concepts focused around major flowpaths.

Answer: We agree that in a sense DRIPs could be considered a part of the stream network especially during a high flow conditions. However, it is in our eyes essential to consider not only the hydrological definition but also the biogeochemical definition. And for the latter, the most important property of DRIPs is the highly organic substrate, and the generally high water tables. This is something that is not typical for stream channels and therefore we would argue that they are a unique landscape feature.

In essence I support this manuscript but ask for a substantial revision of the discussion section and a better justification of field and statistical methods.

Answer: Thank you for the constructive comments.

Specific comments

Abstract:

L13: This sentence is potentially misleading. What is meant by chemical variability in the riparian boreal forest? The linkage of riparian groundwater as described beforehand forest is not clear.

Answer: We will reformulate the sentence.

L20: The pairing of hydrological connection and groundwater condition cannot be understood here. Can you find a more telling factor name?

Answer: The hydrology between DRIP and non-DRIP is mostly conceptualized as different groundwater flow principles based on the water table characteristics. However we will re-evaluate how we can line up the definition and factor names better.

L20: The water provided by DRIPS – is that surface water discharging from the DRIP or groundwater within the DRIP?

Answer: Since the term DRIP covers both subsurface flows, stream-like flows and other types of confluences, we prefer to not specify whether DRIP water is surface water of groundwater. The section 2.2 clarifies this, but within the abstract this might not suffice.

L23: "chemically more stable" may be misleading. Do you mean spatially and/or temporally homogeneous?

Answer: Yes, we will reformulate this.

Introduction

P2L1: While I in general like such comparisons the idea of headwaters as "capillaries" was not immediately clear to me.

Answer: We will further clarify the sentence

P3L10-13: These two sentences need some references.

Answer: We will expand the references in this section.

Methods

Fig. 2: The catchment delineation is somewhat distracting. I suggest to limit the catchment boundaries to the catchment with studied DRIPs/ nonDRIPs. What determines the catchment outlet? A gauging station? Maybe show them to make clear that this was not completely arbitrarily chosen.

Answer: The catchments are determined by gauging stations, and we will indicate

them in the figure. We will reduce the amount of catchment boundaries to clear up the figure.

P5L6: TWI units are not [ha] – what do you mean by "topographic wetness index (2 ha threshold)"?

Answer: We aimed to specify that we used, similar to stream initiation thresholds, a 2 ha lower limit for a DRIP to be predicted as a DRIP. We will clarify this in the text.

P6L1: Can you specify the wells in terms of diameter and material? What is the meaning of a groundwater sample taken in a fully screened well? Do you expect this to be a representative sample from all depths or rather a sample from the most conductive depth? In the latter case the depths where most water is coming from may change over time with changing groundwater levels. I would like to see a critical evaluation of your sampling design and drawbacks (assets?) of the chosen methods! The methods chapter is likely not the right place.

Answer: In the discussion section we will evaluate our sampling design, and put into context with studies that consider samples from different depths (e.g. the DSL and RIM model in case of the Krycklan catchment). We will further specify the well properties in the methods section. The fully screened wells give a representative sample of the water that is laterally transported at the time of sampling, relative to the hydraulic conductivity of each soil layer which the well is in contact with. In our view this is the most representative water sample of what is drained into the stream in terms of shallow groundwater contributions. Given the exponentially decaying hydraulic conductivity profile in these soils, it is likely that our shallow well network provides a representation of the subsurface water that ends up in the headwater streams. Perhaps further down the stream network, at higher order streams, we have to consider a considerable contribution of deep groundwater flow that originates from deeper as the typical well depth of our current well network.

Results

P7L4: I don't understand why you jump in directly with that statement in the second sentence already.

Answer: The sentence aims to explain that the position relative to the stream is most explanatory, but the differences observed are not as great as the contrasts observed between DRIP and non-DRIP water. This means that in our approach we observed the chemical function of the riparian zone in a lateral context (chemical change from the upland wells to the riparian wells), but in the spatial context the DRIP/non-DRIP distinction seems to be more informative given the larger differences that we observed.

P7L26: Did you expect more distinct differences? Keep in mind that the pH is a logarithm of concentrations and small changes can mean a lot compared to DOC and EC.

Answer: In earlier work the soil pH in DRIPs were found to be distinctly higher as non-DRIPs, which was expected to be reflected in the groundwater pH (Kuglerová et al., 2014). We were aware of the pH scales sensitivity but expected perhaps more distinct differences given the field-based observations of DRIP and non-DRIP riparian areas.

P8L16: When you talk about seasonality you mean factor TIME, right? It would be helpful to stick to those factor names and provide it in brackets if other words such as seasonality are used.

Answer: We will go through the result section to ensure this terminology is used consistently

Fig. 4: I suggest that you indicate if the mean is significantly differing in individual panels. That would improve readability here. Same is true for Fig. 3 and 5

Answer: Although we agree that this would improve the figure in general, it is in combination with our statistical approach likely to cause more confusion than clarification. The significant differences of these individual sets might not be entirely representing the statistical outcome of the LMM we have used, since that tests to what degree variability in the predictor can be explained by the explanatory factors. As such, the significant differences of the boxplots is maybe confusing because we deliberately tried to avoid this type of statistic. Moreover the box plots show medians and p-values would be computed by means. Although we do provide a p-value for the Figure 3 box plots, this is a general statistic that does not do justice to the complexity of the dataset. Figures 3-5 are mostly aiming to show the elapse in variability over space and time, rather than proving significant differences at the individual plot or season level.

Discussion

The discussion makes strong links of groundwater spatial variability to stream chemistry temporal variability. But this is not what was shown in the results. I miss a discussion of why groundwater quality was as it was measured and presented above. All the discussion is rather focusing on implications.

Answer: We will expand the first paragraph to further explain the results and shorten the implications for stream water quality.

P9L23 to P10L3: I have problems following the argumentation here. Why does the contrasting chemistry of DRIPS and non-DRIPS explains why pre-event water is quickly mobilizes? Do we really need DRIPS and non-DRIPS to explain temporal variability of stream chemistry within an event? That is also covered by vertical chemical heterogeneity (taking Seibert's RIM for instance).

Answer: We argue that the vertical chemical heterogeneity is still valid, but that in a horizontal plane the hydrological conditions (or storage state) of the riparian zone are dominating the way this vertical chemical heterogeneity is translated to lateral contributions to streams. For DRIPs the mobilization can be extremely fast since there is no unsaturated zone to store more water. In contrast, non-DRIPs are presumably delayed in response due to the vertical infiltration and rise of groundwater tables before a contribution to the stream is initiated. In combination with previous comments we will elaborate on the comparison to the DSL and RIM concepts in the discussion.

P10L26: The generalization (that totally make sense) of your findings to the larger scale of the Kryckland catchment is a selling point. I suggest to base the statement made here on DRIPS in catchments on a sound and reproducible analysis and not on a personal communication.

Answer: The statement is based on a reproducible analysis which we will provide in the supporting material.

---

## Author Comment (AC2) · 16 Oct 2019

I have reviewed the draft manuscript: 'Are hydrological pathways and variability in groundwater chemistry linked in the riparian boreal forest?' submitted by Ploum et al for possible publication in HESS. I like the general premise of this study, eg that preferential flowpaths from hillslopes through riparian zones need to be better considered when characterizing the baseflow controls on stream chemistry and dissolved organic carbon availability (DOC). Too many riparian studies are based around 'uniformed' or random piezometer transect designs, and without the hydrological flow context, the groundwater chemistry data are difficult to interpret. I do think this material is appropriate for HESS, though the paper could benefit from a change in emphasis and additional heat tracing data that it seems the authors may have already collected.

Answer: Thank you for reviewing the manuscript and providing constructive suggestions to improve the manuscript. We will repeat the comments and provide the answers one by one.

Currently, the primary question addressed by the study is posed as: 'Are hydrological pathways and variability in groundwater chemistry linked in the riparian boreal forest?' There have been many studies to document strong variance in groundwater chemistry as controlled by varied advective flowpaths, too numerous to list here. The current study by Ploum and all is unique from many of these as the preferential flowpaths in question source varied dissolved constituents to headwater streams. I suggest the authors refocus the main question to something like: 'Do DRIP's represent preferential conduits of DOC-rich groundwater to headwater streams in a boreal forest?' Or something at least more specific to this study than 'variability in groundwater chemistry'. It seems that the most compelling data presented in this study show the 'DRIPs' in this boreal headwater system are enriched in DOC, which presumably results in higher flux of DOC to the channel via preferential shallow groundwater discharge compared to more diffuse flow through till (though hydrological fluxes are not actually measured or inferred here). Further, the authors document interesting temporal trends in DOC and SpC, the latter being much less meaningful without additional chemical characterization.

Answer: We agree with referee #2 that the most relevant aspect of the DRIP concept is the implications for streams, rather than groundwater chemistry on its own. However, in the line of previous work regarding their riparian function (Kuglerová et al., 2014), the hydrological contributions of DRIPs (Leach et al., 2017; Ploum et al., 2018), and the implications for stream chemistry (Lupon et al., 2019), the aim of this contribution is the characterization of DRIP groundwater chemistry relative to non-DRIP riparian zones. We wanted to represent this specific aim in the title and research question,

but as referee #2 understandably remarks the term groundwater chemistry is perhaps too broad. By emphasizing the riparian boreal forest setting, we attempted to steer towards DOC related chemistry however we realize this is not necessarily evident for all readers. We will reconsider the terminology of our title and research question to represent the study outcomes.

In general, I feel the LMM statistical analysis was appropriate for assessing DRIP/non DRIP well DOC, pH, and EC. The results between these binary classifications are interesting, though for all the effort on well installation and sample collection some basic hydrological data seem missing. Where lateral gradients measured between wells and the stream? Were any hydraulic conductivity tests performed? Do we have any idea of groundwater flow rate/flux to the stream from DRIP vs non-DRIP zones? This is a flowpath-based study but the reader is left without any real flow-based hydrogeological information. The addition of some basic quantitative hydrogeological data, and perhaps some additional measured parameters such as dissolved oxygen, could have nicely increased the impact of this study.

Answer: We agree that quantitative data on groundwater hydrology would greatly benefit the study. However currently we have no comprehensive groundwater flux data to quantify flow from DRIPs and non-DRIPs. A few hydraulic conductivity tests were performed under summer low flow conditions in 4 DRIP wells, but we did not deem this representative to use in the study. Instead we based our concept on the assumption that topography and hydraulic gradients are similar in the boreal headwaters. Previous work has shown that continuous saturated conditions occur in the DRIPs and that topographic gradients are low (Kuglerová et al., 2014). Also DEM based flow accumulation model and in-stream measurements indicate that DRIPs provide disproportionally large water contributions, but they remain difficult to quantify and their detectability varies throughout various events (Leach et al., 2017; Lupon et al., 2019; Ploum et al., 2018). We consider the quantification of groundwater fluxes as an interesting future research topic, but it is beyond the scope of this contribution and currently there is no

comprehensive dataset available to perform such a study. We will however elaborate on the well installations in the methods and elaborate on the hydrogeological setting.

Without any evaluation of piezometer water age (eg dissolved gas-, isotope-based) the discussion of old vs new water contributions is highly speculative and should be scaled back. I really like the concept you put forth of DRIPs as drivers of younger water fractions in streams where low permeability soils dominate (eg tills), though it is difficult to support this using EC as the primary parameter. Despite these criticisms I do think this work could make an important addition to HESS after some revision considering the major comments below and that of the other reviewers.

Answer: We agree that our young-old water discussion is speculative. In combination with comments of referee #1 we agree to scale back this section of the discussion. Instead we suggest an evaluation of the DRIP/non-DRIP concept with existing concepts of groundwater-surface water interactions (DSL and RIM model (Ledesma et al., 2015; Seibert et al., 2009)).

The statement is made toward the end of the Introduction: 'However, in order to determine whether DRIPs matter for stream biogeochemistry, chemical characterization of the discharging groundwater is needed.' Yes, but this is only half of the equation, the other being the flux of groundwater to the stream which was not evaluated here in any way. It seems at least some data specific to groundwater discharge was collected in these streams and presented by Ploum et al 2018. Where many of the DRIPs instrumented here with piezometers identified in the stream as preferential discharge points using heat tracing? If so that data could be briefly included here with an additional figure, and go a long way to convincing the reader that the wet topographic low points mapped here as DRIPs are actual preferential flowpaths from the hillslope to the stream. Without any such thermal or hydrological gradient/permeability data it is difficult to accept with confidence that the DRIPs mapped here are actual preferential flow zones, compared to the surrounding soils. To be clear I think that the hydrogeologic interpretation of DRIPs made by the authors is likely generally correct, particularly after reading/watching Ploum et al 2018, but the current paper would benefit greatly from some groundwater flow-based data.

Answer: We agree with referee #2 that anomalies in groundwater chemistry without hydrological fluxes have limited meaning for streams. However for this study we argue that there is already sufficient support to assume that the studied DRIPs have important implications for stream chemistry, which would be undetectable if hydrological fluxes were no different than non-DRIP riparian zones. The earlier mentioned work on our DRIP sites showed that the DRIPs provide the majority of the lateral fluxes to the stream using thermal and isotope stream signatures (Leach et al., 2017). Although this study also demonstrated that the fluxes are difficult to match with contributing areas of the DRIPs, biogeochemically the DRIPs alter streams such that observable differences have been reported in gas fluxes as a result of stream processes (Lupon et al., 2019). This leads us to believe that, although we currently have no reported hydrological fluxes of all the studied DRIPs, they can be considered as the dominant lateral hydrological fluxes to the stream. In addition, the hydrogeological information mentioned above will further support that the DRIPs are major hydrological contributions to streams.

I list several more major comments below (I realize some are a bit redundant with this narrative) followed by some more minor text suggestions:

1. A main conclusion of this work is stated as: 'We concluded that hydrological pathways and spatial variability in groundwater are linked, and that DRIPs are important control points in the boreal landscape.' Can you build on this statement in the abstract to be more specific to your study? Near stream shallow chemical variability has long been linked to flowpaths. Perhaps comment more specifically in the abstract regarding the spatial variability you observed to set your work apart from previous studies. The Results section discusses some temporal patterns, but I do not see this data reflected in any of the figures or tables. You could develop a figure specific to the interesting temporal patterns, this information is shown somewhat in Table 1 'TIME' variable analysis but could be shown more explicitly. Also, I think the fact that your research indicates

DRIPs our important DOC pathways to the boreal stream corridor is quite important.

Answer: We will emphasize in the abstract the context and the setting to which our findings apply. Given the specific study site and the generic claim we make, we understand that this can be interpreted in various ways. For the second point we want to clarify that the temporal patterns we refer to are the seasonal differences that are reported as TIME in the analysis (spring, summer, autumn). We will clarify this.

2. Under your definition, are DRIPs only driven by surface topography and wetness? There are numerous instances of preferential discharge of groundwater and interflow through the riparian zone through highly permeable sediments that are not correlated with surface topography, and in glacial sediments often occur at local topographic high points (sand and gravel deposits transecting the riparian zone). I agree that local topographic depressions often lead to saturated conditions in the riparian zone, but that is not the same thing as strong hydrologic connectivity to the stream channel, which depends on the combination of lateral gradient and sediment permeability. Previous work by this group (Ploum et al 2018) used heat tracing methodology to locate/confirm preferential discharge of riparian water to the stream channel, which makes sense. However the current work does not seem to tie the definition of DRIP to actual observed high-discharge points, which I think is unfortunate. Not all saturated depressions will be points of preferential discharge to the channel, which strongly depends on soil permeability. Further, according to your statement: 'water in DRIPs travels a longer distance horizontally; in presumably wet, highly permeable, organic rich soil." It seems your definition of DRIP is relatively narrow, and based around forested headwaters similar to where your study has been conducted. I think it would be quite helpful for you to more specifically define 'DRIP' early in the manuscript (in the Introduction), and acknowledge that this definition applies to only a subset of preferential riparian groundwater discharge zones in headwater systems. Your broad definition of DRIP in section 2.2 (eg 'groundwater discharge zone, groundwater hotspots, cryptic wetlands, swales, focused seepage, discrete seepage, springs, upwelling zones, preferential discharge,

ephemeral streams and zero-order streams') does not seem to apply to the functional definition you apply here for shallow lateral flow above the mineral soil horizon, so please be clear on what definition you are using for this study.

Answer: We agree that preferential discharges not necessarily follow surface topography, and that saturation in topographic depressions does not unconditionally infers hydrological connectivity. DRIPs differ from generic saturated near-stream areas by the large contributing area that they drain. The location of DRIPs in this study were predicted using surface topography and wetness, but the condition is that their contributing area is disproportionally larger than the remaining riparian zone. To ensure that not any saturated area is assigned as DRIP, the predicted DRIPs are also field validated by thermal detection using the DTS system and/or visual inspection (Kuglerová et al., 2014; Leach et al., 2017). Altogether, surface topography is therefore an important aspect of our definition of DRIPs, and we realize that preferential flow to a stream is not always represented by that property. We will put our DRIP definition into context of the studied landscape and also account for systems outside the boreal forest. In addition we will clarify and synchronize the definition of DRIPs in the abstract, introduction and method section.

3. It is not clear to me how DRIPs are defined for 'upland' areas. . . do you use the same definition based on topographic depression and wetness index that you use for the riparian zone DRIP areas?

Answer: Yes, the upland wells in the DRIP transects were predicted based on the same criteria as riparian DRIP wells. The exact location of upland wells of DRIPs were then determined in the field starting from the riparian well following the surface topography in order to approximate the most likely hydrological flow paths. The additional hydrogeological information in the methods should clarify this.

4. Abstract L10 and elsewhere: it is somewhat of a misnomer that all riparian flowpaths lead to biogeochemical alteration of discharging water chemistry. Low-carbon mineral

soils and highly preferential flowpaths such as peat pipes and other macropores can yield little alteration in hillslope and deeper groundwater discharge. In fact your findings indicate that DRIPs lead to less net reaction then more diffuse groundwater discharge through the riparian zone: 'Moreover, DRIPs were chemically more stable from the upland area to the stream'. You might check out this commentary for some relevant discussion : (https://onlinelibrary.wiley.com/doi/10.1002/hyp.11153)

Answer: We agree that the generalization might be too broad. We will reformulate this and we thank the referee for providing the reference to develop our discussion.

5. It would seem fully screened wells down to apprx 1.5 m depth would integrate shallow 'DRIP' water and deeper mineral soil water, how did you account for this?

Answer: The wells were drilled until resistance or until a first aquitard was encountered. Given the exponentially decaying hydraulic conductivity profile, we can assume that the majority of the water is DRIP (or non-DRIP) water. We will explain this in more detail in the material and method section.

I think your statements in the Discussion section regarding 'old' and 'young' water are a bit to speculative, as this is essentially only based on EC data, a parameter influenced by a number of flowpath process. It does not seem like any age dating/isotope analysis was performed, so how confident can you be regarding relative water ages/residence times?

Answer: We agree that our young/old water discussion is speculative, and considering other comments we suggest to downscale this section.

Also, you mention the piezometers were installed until they reached a hard layer. Did this depth vary systematically from DRIP to non-DRIP locations? If so you could include that data, as depth to rock/confining layer can also be a strong control on shallow groundwater flowpath chemistry.

Answer: We will elaborate on this in the material and method section. Typically the

drilling was limited by large pebbles and not bedrock. This was around 1.2-1.5 meter depth.

Fig 1: Although we might expect local shallow percolation in non-DRIP near stream zones, groundwater flow is likely dominated by the lateral component toward the stream (in gaining stream reaches), though the discharge magnitude may be reduced compared to preferential discharge points. I suggest you alter the 'vertical groundwater flow' language in the 3rd panel of this Figure, the vertical flow you refer to may instead by non saturated percolation toward the water table, where groundwater flow is predominantly horizontal.

Answer: We agree that non-saturated percolation is a more appropriate term for this process and will change the figure accordingly

Have you measured any vertical head gradients at the wells, and lateral gradients between wells, to support these conceptual diagrams?

Answer: We will provide typical groundwater level timeseries in a broader hydrogeological section in the materials and methods. We don't have current head gradients to present complimentary to the conceptual figure.

Fig 3- The caption could be simplified, you do not need to define DRIPs in the caption as this is done in the text

Answer: We will change the caption

Minor points:

Pg 2: L2 repeat of the word 'landscape', please look for replacement

Answer: We will rephrase the sentence

L3 I am not sure what you mean by 'newly introduced water', can you be more specific?

Answer: We mean event water, we will clarify the sentence

L19 do you have a reference example to associate with: 'Traditionally, streamflow generation has often been assumed to be driven by spatially diffuse groundwater exchange often released at a constant rate.'?

Answer: We will provide references for this statement

Pg 4 L20: could cite here the hydrographs shown in Ploum et al 2018

Answer: We will provide the reference and also elaborate more on the hydrogeological setting

L24: you would not consider the fall period to be 'baseflow' dominated as well or is this just a winter condition in your watershed system?

Answer: We considered baseflow as the flow conditions in winter period given the snow dominated region we performed our study. In autumn low flow conditions can occur but not as long and consistent as during winter due to regular rain events. In winter snow and soil frost inhibit flow in the shallow subsurface and 'true' baseflow conditions occur.

Pg 7 L13: replace 'double as high'

Answer: We will rephrase the sentence

---

## Author Comment (AC3) · 16 Oct 2019

Variability in landscapes is a challenge for understanding how landscapes influence water chemistry in space and time. By developing a sampling design based on a hypothesis about how water is flowing through the riparian zone, this study has provided new insights into the hydrobiogeochemical structures that shape the connection between landscapes and waters. This has practical implications for the design of buffer strips that are widely used in water management. As such this paper can be a valuable contribution to the literature. I think its value would be enhanced if a few points in the paper got further attention.

[Figure]

Answer: Thank you for providing this short comment, we will incorporate the points in our revised manuscript. Below we answer the comments one by one.

One concerns the distinction drawn between DRIPs and the confluences of ephemeral streams on page 5, lines 19-21. The text here was not clear. I think the authors are trying to say that such confluences are not included in their definition of DRIPS since DRIPS do not have clear channels. It would be good if this could be clarified.

Answer: We will clarify this section. We agree that emphemeral stream and DRIPs differ from each other by the presence of a clear channel in the case of emphemeral streams, while this is not the case for DRIPs. In practice it can occur that DRIPs have a small channel-like appearance within the very last meter when it merges with the stream, for example when there is bank height difference.

A second point I suggest that the authors address concerns the discussion of relative contributions from DRIPs and Non-DRIPS to stream chemistry under different flow conditions (page 10, lines 29-34). This part of the discussion talks about the contrasting chemistries coming from DRIPS and non-DRIPS. But the effect of chemical differences in the source waters on stream chemistry depends on the proportion of water coming from the different source waters. Is there some assumption underlying this part of the discussion about how much water comes from DRIPs relative to non-DRIPS during high and low flow conditions? Clarification of that would help make the points in this part of the discussion more persuasive.

Answer: Thank you for moving this discussion further. In terms of flow conditions, the speculations were related to high flow conditions during a hydrological event. In the sentences before this section, we presented that 57% of the Krycklan catchment drains through DRIPs, which connect to only 12% of the stream network. These numbers happen to be similar to the distribution of DRIPs in the studied headwater catchment of Krycklan where our well network is located (Leach et al., 2017). However, since we have reported earlier that detectability of DRIPs changes throughout events and

seasons (Ploum et al., 2018), this assumption can be argued against and it is very likely that DRIP/non-DRIP contributions vary over time.

In the lines 29-34 we speculated that the distribution of the DRIPs in other headwaters of the Krycklan catchment might differ, which would lead those headwaters to have different young/old water fractions. This suggestion was based on the idea that due the high storage state of DRIPs, their activation is much faster compared to non-DRIP riparian zones, which lead to our suggestion that young fractions are mobilized quickly, contrary to non-DRIP areas. Moreover, this line of thought had the underlying assumption that the chemical characteristic of DRIPs and non-DRIPs would stay the same (low EC, high DOC for DRIPs and high EC, low DOC for non-DRIPs), but that the landscape organization might be different in different headwaters (more or less DRIPs). It is very likely that this chemical contrast between DRIP and non-DRIPs is not consistent over the entire catchment. Although this is an interesting discussion, we suggest that based on the many underlying assumptions and the comments of the referees #1 and #2, we will scale down this section of the discussion. Instead we will more focus on the context of DRIPs in groundwater flow to streams in boreal till landscapes, considering the RIM and DSL concepts (Ledesma et al., 2015; Seibert et al., 2009).

Furthermore, if the DRIPS do not include the confluences of intermittent streams with the perennial stream channel, it would be important to mention what these ephemeral streams are doing to contribute to the high-flow stream chemistry being talked about in the discussion.

Answer: We agree that intermittent streams should be included when evaluating high-flow stream chemistry. A clean channel-like feature might respond faster to hydrological inputs compared to DRIPs. That being said, relative to non-DRIP riparian hillslopes, the responsiveness of shallow subsurface runoff generation of DRIPs possibly falls in the same order of magnitude as intermittent streams. Similar to the comment above, we suggest that due to the speculation in our discussion we scale down this part of the discussion. In the DRIP definition in the method section we will clarify how DRIPs

relate to intermittent streams.

One final question, the concept of a "DRIP initiation threshold" is mentioned on page10 line 35, but a definition of what this means is not given. Please explain the term.

Answer:The initiation threshold is the TWI threshold referred to in line 6 page 5 (2ha). This is a subjective threshold, but further supported by the flow accumulation model of the stream channel. We will clarify this in the text.

———————————————

---

## Author Response (AR1)

**Dear Editor,**

Thank you for considering this revised manuscript. We provide here a summary of the major changes in form a bullet-point list.

- We have specified the title, research question and overall emphasis towards DOC in riparian groundwater, instead of the earlier used "spatial variability in groundwater".
- We have revised the methods section such that it contains more hydrogeological background and information about the well infrastructure. We added in the supplementary materials an example of groundwater level data to demonstrate how DRIP and non-DRIP differ from each other.
- We have clarified the definition of DRIPs in the introduction, methods and discussion, and elaborated on the differences to the rest of the riparian zone, and ephemeral streams.
- We rewrote the discussion and removed the speculations about implications for stream chemistry. Instead we have focused on contextualizing our findings to other work related to riparian groundwater.

In the section below we answer the review comments one by one in red, and provide the manuscript with tracked changes

**Page numbers and lines refer to the revised manuscript without tracked changes.**

**General comment**

This is a concise paper that compares groundwater chemistry in sub-catchments of the Kryckland observatory and asks questions about chemical variability in relation to hydrological pathway. The paper is written in a clear and concise way with not much to complain about in the introduction and results. The paper clearly shows and describes the variability of groundwater chemistry in relation to its hydro- logical activity. This is a very relevant result that is worth being published in HESS.

Answer: Thank you for evaluating the manuscript and the constructive commentary. Below we will present responses in red to each point that is addressed.

However, I have some point of criticism that should be addressed: Groundwater was sampled with fully screened wells which means that there is an unknown integration of water quality over depth potentially changing between sampling campaigns. The authors should carefully discuss the pros and cons of this approach.

Answer: We agree with Referee #1. We extended section 2.2, for example:

P5, L20: "We assumed that the water sampled from the well is a weighted average of the phreatic aquifer, down to the depth of the well. Given the exponentially decaying hydraulic conductivity with depth, this assumption would imply that, under fully saturated conditions, the majority of the water is therefore from the upper soil layers, referred to as the dominant source layer (Ledesma et al., 2015)."

**and added a paragraph in section 4 (Discussion P10, L39, see below).**

Later on in the text there seems to be a not-outspoken assumption of vertical homogeneity of groundwater quality and a high weight on lateral heterogeneity. This needs to be made more clear and concise as this new DRIP-concept seems to stand against the former RIM- (Seibert et al. 2009) and DSL-concepts (Ledesma et al. 2015) that were derived from the same study site.

Answer: We agree that further elaboration was needed, in combination with the previous comment, regarding the vertical heterogeneity in the groundwater quality and the associated RIM and DSL concepts. Our study could be considered complimentary to the existing literature, rather than opposing. To clarify this we changed the introduction, for example:

P2, L21: "Some models, such as the RIM model and DSL concept, have considered the vertical heterogeneity in riparian groundwater fluxes to boreal streams (Ledesma et al., 2015; Seibert et al., 2009)."

P3, L16: "Also the RIM model has provided a framework to infer groundwater chemistry profiles from stream chemistry (Seibert et al., 2009)."

In the discussion we have added a section starting P10, L31, in which we address the following on L39:

"In that way, our study can be contextualized as an approach that potentially allows characterization of control points in the landscape with use minimal information. The relative contributions and biogeochemical characteristics of DRIPs and non-DRIP riparian zones in the longitudinal dimension, can potentially be combined with models that specify vertical profiles of groundwater chemistry, such as the RIM model (Seibert et al., 2009). As such we can identify within the riparian zone which parts exert a large control on stream water quality and quantity. "

The authors used linear mixed-effect models (LMM) to analyze spatiotemporal patterns of groundwater chemistry. There should be some effort in the text to justify the choice of this method and also on the choice of predictors. One could argue that the authors did not look hard enough to include other predictors (e.g. local TWI) but instead chose a method which can handle random effects from unaccounted factors.

Answer: The use of LMM was justified by the setup of the groundwater well network. Given that this is, to our knowledge, the first groundwater chemistry database that has been designed and compiled a priori based on three factors (hydrological pathways, distance to stream and season), we deem those factors as our only true testable factors. Since the study sites were selected partly based on TWI, TWI and DRIP vs. non-DRIP are inherently correlated. Given the large improvement of the model by the inclusion of random factors which are describing spatial dependency of the samples, we discussed that there is a large part of spatial variability in groundwater chemistry left to be explained by factors that are not included by our study. We deliberately attempted to explain groundwater chemistry in a simple manner with generic and a priory set predictors, and explaining variability by unique properties of the study sites would impair any upscaling of our findings.

Finally, I have a major concern with the discussion section that is rather an implication section centered on the question what the varying groundwater quality would mean for stream chemistry. However, this was not part of the study design. So the discussion should rather be focused around the question, why the measured groundwater quality was different in different parts of the riparian zone.

Answer: We agree that the discussion was rather speculative and conceptual. We have rewritten the discussion section (P9 to P11), with the aim to have a balanced interpretation of our findings with other groundwater-based studies, and reduced speculations regarding implications for streams.

What came into my mind here: Aren't DRIPs just an extension of the fractal stream network into the catchment. DRIPs are along topographic depressions funneling water flow in the same way as the stream network. The major difference is that they are not (permanent) pathways of surface flow but rather pathways of shallow subsurface flow. Maybe I missed that connection in earlier studies that were cited here. I however very much like the idea to refine the view on riparian zone by that type of concepts focused around major flowpaths.

Answer: We agree that in a sense DRIPs could be considered a part of the stream network especially during a high flow conditions. However, it is in our eyes essential to consider not only the hydrological definition but also the biogeochemical definition. And for the latter, the most important property of DRIPs is the highly organic substrate, and the generally high water tables. This is something that is not typical for stream channels and therefore we would argue that they are a unique landscape feature.

In essence I support this manuscript but ask for a substantial revision of the discussion section and a better justification of field and statistical methods.

Answer: Thank you for the constructive comments.

Specific comments Abstract:

L13: This sentence is potentially misleading. What is meant by chemical variability in the riparian boreal forest? The linkage of riparian groundwater as described beforehand forest is not clear.

Answer: The sentence was:

"Given the important chemical function of the riparian zone, we therefore ask the question: are hydrological pathways and chemical variability linked in the riparian boreal forest?"

We changed this section and now the related sentence is formulated (P1, L18):

"We therefore ask the question: are DOC concentrations in riparian groundwater linked to hydrological pathways in the boreal forest?"

L20: The pairing of hydrological connection and groundwater condition cannot be understood here. Can you find a more telling factor name?

Answer: We changed the factor name to HP, hydrological pathways (P6, L24).

L20: The water provided by DRIPS – is that surface water discharging from the DRIP or groundwater within the DRIP?

Answer: Combined with the short comment we realized our definition of DRIPs was not clear and raised confusion. DRIPs are discharging groundwater from the shallow subsurface. During events the watertable can rise above the surface and have a surface water like appearance. As such they are not surface waters, or part of the stream channel. We have included a picture in the supplementary materials to clarify what DRIPs look like in the landscape (Fig. S2).

L23: "chemically more stable" may be misleading. Do you mean spatially and/or temporally homogeneous?

Answer: We have adopted the suggestion.

Introduction

P2L1: While I in general like such comparisons the idea of headwaters as "capillaries" was not immediately clear to me.

Answer: We rephrased the sentence to:

"Headwater streams can be seen as the capillaries of the landscape: although small in appearance, collectively they make up the majority of a stream network."

P3L10-13: These two sentences need some references.

Answer: These sentences were changed to:

"Electrical conductivity (EC) can be used as a proxy for the ionic strength, or total amount of dissolved ions in water (Corwin and Lesch, 2005). Water contact time with minerals and weathering processes are important factors determining EC (Saarenketo, 1998), with increasing EC indicating longer interactions (Hayashi, 2004; Peralta-Tapia et al., 2015). "

Methods

Fig. 2: The catchment delineation is somewhat distracting. I suggest to limit the catchment boundaries to the catchment with studied DRIPs/ nonDRIPs. What determines the catchment outlet? A gauging station? Maybe show them to make clear that this was not completely arbitrarily chosen.

Answer: We have updated the figure. The gauging station at the outlet was indicated and catchment boundaries removed.

P5L6: TWI units are not [ha] – what do you mean by "topographic wetness index (2 ha threshold)"?

Answer: This section was changed to:

P5,L7: "Discrete Riparian Inflow Points (DRIPs) were selected by considering wet areas, based on a topographic wetness index, and selecting large step changes in catchment area along stream networks using flow accumulation algorithms (Ågren et al., 2014; Beven and Kirkby, 1979; O'Callaghan and Mark, 1984). The DRIPs (n = 10) were selected with contributing upslope area varying from 0.6 to 7.7 ha, with a mean contributing area of 2.7 ha."

P6L1: Can you specify the wells in terms of diameter and material? What is the meaning of a groundwater sample taken in a fully screened well? Do you expect this to be a representative sample from all depths or rather a sample from the most conductive depth? In the latter case the depths where most water is coming from may change over time with changing groundwater levels. I would like to see a critical evaluation of your sampling design and drawbacks (assets?) of the chosen methods! The methods chapter is likely not the right place.

Answer: We have changed section 2.2 to include more details about the well installations P5L15-P6L3. In combination with the earlier suggestion to change the discussion, we included a section covering different approaches (P10L31).

Results

P7L4: I don't understand why you jump in directly with that statement in the second sentence already.

Answer: We have restructured the paragraph (P7L3).

P7L26: Did you expect more distinct differences? Keep in mind that the pH is a logarithm of concentrations and small changes can mean a lot compared to DOC and EC.

Answer: In earlier work the soil pH in DRIPs were found to be distinctly higher as non-DRIPs, which was expected to be reflected in the groundwater pH (Kuglerová et al., 2014). We were aware of the pH scales sensitivity but expected perhaps more distinct differences given the field-based observations of DRIP and non-DRIP riparian areas.

P8L16: When you talk about seasonality you mean factor TIME, right? It would be helpful to stick to those factor names and provide it in brackets if other words such as seasonality are used.

Answer: We have added TIME in brackets where we used seasonality or other terms that refer to the TIME factor in section 3.

Fig. 4: I suggest that you indicate if the mean is significantly differing in individual panels. That would improve readability here. Same is true for Fig. 3 and 5

Answer: The box plots show medians and p-values would be computed by means. Although we do provide a p-value for the Figure 3 box plots, this is a general statistic that does not do justice to the complexity of the dataset. Figures 3-5 are mostly aiming to show the elapse in variability over space and time, rather than proving significant differences at the individual plot or season level.

**Discussion**

The discussion makes strong links of groundwater spatial variability to stream chemistry temporal variability. But this is not what was shown in the results. I miss a discussion of why groundwater quality was as it was measured and presented above. All the discussion is rather focusing on implications.

**Answer: We have rewritten the discussion section and reduced the speculations regarding stream implications (P9-P11).**

P9L23 to P10L3: I have problems following the argumentation here. Why does the contrasting chemistry of DRIPS and non-DRIPS explains why pre-event water is quickly mobilizes? Do we really need DRIPS and non-DRIPS to explain temporal variability of stream chemistry within an event? That is also covered by vertical chemical heterogeneity (taking Seibert's RIM for instance).

Answer: We argued that the vertical chemical heterogeneity is still valid, but that in a horizontal plane the hydrological conditions (or storage state) of the riparian zone are dominating the way this vertical chemical heterogeneity is translated to lateral contributions to streams. For DRIPs the mobilization can be extremely fast since there is no unsaturated zone to store more water. In contrast, non-DRIPs are presumably delayed in response due to the vertical infiltration and rise of groundwater tables before a contribution to the stream is initiated. In combination with previous comments we have included more explicitly the DSL and RIM concepts in the discussion.

P10L26: The generalization (that totally make sense) of your findings to the larger scale of the Kryckland catchment is a selling point. I suggest to base the statement made here on DRIPS in catchments on a sound and reproducible analysis and not on a personal communication.

Answer: The statement was based on a reproducible analysis which we provided in the supporting material.

I have reviewed the draft manuscript: 'Are hydrological pathways and variability in groundwater chemistry linked in the riparian boreal forest?' submitted by Ploum et al for possible publication in HESS. I like the general premise of this study, eg that preferential flowpaths from hillslopes through riparian zones need to be better considered when characterizing the baseflow controls on stream chemistry and dissolved organic carbon availability (DOC). Too many riparian studies are based around 'uniformed' or random piezometer transect designs, and without the hydrological flow context, the groundwater chemistry data are difficult to interpret. I do think this material is appropriate for HESS, though the paper could benefit from a change in emphasis and additional heat tracing data that it seems the authors may have already collected.

**Answer: Thank you for reviewing the manuscript and providing constructive suggestions to improve the manuscript.**

Currently, the primary question addressed by the study is posed as: 'Are hydrological pathways and variability in groundwater chemistry linked in the riparian boreal forest?' There have been many studies to document strong variance in groundwater chemistry as controlled by varied advective flowpaths, too numerous to list here. The current study by Ploum and all is unique from many of these as the preferential flowpaths in question source varied dissolved constituents to headwater streams. I suggest the authors refocus the main question to something like: 'Do DRIP's represent preferential conduits of DOC-rich groundwater to headwater streams in a boreal forest?' Or something at least more specific to this study than 'variability in groundwater chemistry'. It seems that the most compelling data presented in this study show the 'DRIPs' in this boreal headwater system are enriched in DOC, which presumably results in higher flux of DOC to the channel via preferential shallow groundwater discharge compared to more diffuse flow through till (though hydrological fluxes are not actually measured or inferred here). Further, the authors document interesting temporal trends in DOC and SpC, the latter being much less meaningful without additional chemical characterization.

**Answer: We have changed the title and research question to "Are DOC concentrations in riparian groundwater linked to hydrological pathways in the boreal forest?"**

In general, I feel the LMM statistical analysis was appropriate for assessing DRIP/non DRIP well DOC, pH, and EC. The results between these binary classifications are interesting, though for all the effort on well installation and sample collection some basic hydrological data seem missing. Where lateral gradients measured between wells and the stream? Were any hydraulic conductivity tests performed? Do we have any idea of groundwater flow rate/flux to the stream from DRIP vs non-DRIP zones? This is a flowpath-based study but the reader is left without any real flow-based hydrogeological information. The addition of some basic quantitative hydrogeological data, and perhaps some additional measured parameters such as dissolved oxygen, could have nicely increased the impact of this study.

Answer: We had no comprehensive groundwater flux data to quantify flow from DRIPs and non-DRIPs. A few hydraulic conductivity tests were performed under summer low flow conditions in 4 DRIP wells, but we did not deem this representative to use in the study. Instead we have provided more hydrological background information in section 2.2 (P5L15) and Figure S1.

In addition to that, previous work has shown that continuous saturated conditions occur in the DRIPs and that topographic gradients are low (Kuglerová et al., 2014). Also DEM based flow accumulation

model and in-stream measurements indicate that DRIPs provide disproportionally large water contributions, but they remain difficult to quantify and their detectability varies throughout various events (Leach et al., 2017; Lupon et al., 2019; Ploum et al., 2018).

Without any evaluation of piezometer water age (eg dissolved gas-, isotope-based) the discussion of old vs new water contributions is highly speculative and should be scaled back. I really like the concept you put forth of DRIPs as drivers of younger water fractions in streams where low permeability soils dominate (eg tills), though it is difficult to support this using EC as the primary parameter. Despite these criticisms I do think this work could make an important addition to HESS after some revision considering the major comments below and that of the other reviewers.

**Answer: We agree that our young-old water discussion was speculative and this part of the discussion is replaced by comparisons to existing riparian groundwater concepts.**

The statement is made toward the end of the Introduction: 'However, in order to determine whether DRIPs matter for stream biogeochemistry, chemical characterization of the discharging groundwater is needed.' Yes, but this is only half of the equation, the other being the flux of groundwater to the stream which was not evaluated here in any way. It seems at least some data specific to groundwater discharge was collected in these streams and presented by Ploum et al 2018. Where many of the DRIPs instrumented here with piezometers identified in the stream as preferential discharge points using heat tracing? If so that data could be briefly included here with an additional figure, and go a long way to convincing the reader that the wet topographic low points mapped here as DRIPs are actual preferential flowpaths from the hillslope to the stream. Without any such thermal or hydrological gradient/permeability data it is difficult to accept with confidence that the DRIPs mapped here are actual preferential flow zones, compared to the surrounding soils. To be clear I think that the hydrogeologic interpretation of DRIPs made by the authors is likely generally correct, particularly after reading/watching Ploum et al 2018, but the current paper would benefit greatly from some groundwater flow-based data.

Answer: We agree with referee #2 that anomalies in groundwater chemistry without hydrological fluxes have limited meaning for streams. However for this study we argue that there is already sufficient support to assume that the studied DRIPs have important implications for stream chemistry, which would be undetectable if hydrological fluxes were no different than non-DRIP riparian zones. The earlier work on our DRIP sites showed that the DRIPs provide the majority of the lateral fluxes to the stream using thermal and isotope stream signatures (Leach et al., 2017). Although this study also demonstrated that the fluxes are difficult to match with contributing areas of the DRIPs, biogeochemically the DRIPs alter streams such that observable differences have been reported in gas fluxes as a result of stream processes (Lupon et al., 2019). This leads us to believe that, although we currently have no reported hydrological fluxes of all the studied DRIPs, they can be considered as the dominant lateral hydrological fluxes to the stream. We have added exemplar groundwater level data in Fig S1.

I list several more major comments below (I realize some are a bit redundant with this narrative) followed by some more minor text suggestions:

1. A main conclusion of this work is stated as: 'We concluded that hydrological pathways and spatial variability in groundwater are linked, and that DRIPs are important control points in the boreal landscape.' Can you build on this statement in the abstract to be more specific to your study? Near stream shallow chemical variability has long been linked to flowpaths. Perhaps comment more specifically in the abstract regarding the spatial variability you observed to set your work apart from previous studies. The Results section discusses some temporal patterns, but I do not see this data reflected in any of the figures or tables. You could develop a figure specific to the interesting temporal patterns, this information is shown somewhat in Table 1 'TIME' variable analysis but could be shown more explicitly. Also, I think the fact that your research indicates DRIPs our important DOC pathways to the boreal stream corridor is quite important.

Answer: We have specified our research question towards DOC, as that is the major component of our analysis and our interpretations. The temporal patterns referred to the differences between seasons, which we added as a discussion section starting P9L14.

2. Under your definition, are DRIPs only driven by surface topography and wetness? There are numerous instances of preferential discharge of groundwater and interflow through the riparian zone through highly permeable sediments that are not correlated with surface topography, and in glacial sediments often occur at local topographic high points (sand and gravel deposits transecting the riparian zone). I agree that local topographic depressions often lead to saturated conditions in the riparian zone, but that is not the same thing as strong hydrologic connectivity to the stream channel, which depends on the combination of lateral gradient and sediment permeability. Previous work by this group (Ploum et al 2018) used heat tracing methodology to locate/confirm preferential discharge of riparian water to the stream channel, which makes sense. However the current work does not seem to tie the definition of DRIP to actual observed high-discharge points, which I think is unfortunate. Not all saturated depressions will be points of preferential discharge to the channel, which strongly depends on soil permeability. Further, according to your statement: 'water in DRIPs travels a longer distance horizontally; in presumably wet, highly permeable, organic rich soil." It seems your definition of DRIP is relatively narrow, and based around forested headwaters similar to where your study has been conducted. I think it would be quite helpful for you to more specifically define 'DRIP' early in the manuscript (in the Introduction), and acknowledge that this definition applies to only a subset of preferential riparian groundwater discharge zones in headwater systems. Your broad definition of DRIP in section 2.2 (eg 'groundwater discharge zone, groundwater hotspots, cryptic wetlands, swales, focused seepage, discrete seepage, springs, upwelling zones, preferential discharge, ephemeral streams and zero-order streams') does not seem to apply to the functional definition you apply here for shallow lateral flow above the mineral soil horizon, so please be clear on what definition you are using for this study.

Answer: We have removed the DRIP definition section and refer to DRIPs from the introduction P3L5 onwards. In section 2.2 we have added data that clarifies that the difference in contributing area is a major property of DRIPs that makes is distinctly different from other saturated riparian areas that may not have large contributing areas (P5L7). We have also clarified in the discussion (P9L1) that DRIPs can be seen as a landscape feature that is distinctly different from riparian hillslopes that have been studied in this study area, and that DRIPs do not fit in the definition of ephemeral streams or other features in the existing literature that refer to specific processes.

3. It is not clear to me how DRIPs are defined for 'upland' areas. . . do you use the same definition based on topographic depression and wetness index that you use for the riparian zone DRIP areas?

Answer: Yes, the upland wells in the DRIP transects were predicted based on the same criteria as riparian DRIP wells. The exact location of upland wells of DRIPs were then determined in the field starting from the riparian well following the surface topography in order to approximate the most likely hydrological flow paths.

4. Abstract L10 and elsewhere: it is somewhat of a misnomer that all riparian flowpaths lead to biogeochemical alteration of discharging water chemistry. Low-carbon mineral soils and highly preferential flowpaths such as peat pipes and other macropores can yield little alteration in hillslope and deeper groundwater discharge. In fact your findings indicate that DRIPs lead to less net reaction then more diffuse groundwater discharge through the riparian zone: 'Moreover, DRIPs were chemically more stable from the upland area to the stream'. You might check out this commentary for some relevant discussion : (https://onlinelibrary.wiley.com/doi/10.1002/hyp.11153)

Answer: We agree that the generalization was too broad and we changed this to (P1,L10):

"Here wet, carbon-rich soils can change water chemistry before it enters the stream. In the boreal forest, the riparian zone plays an especially important role in the export of dissolved organic carbon (DOC) to streams."

In addition we have emphasized that the function of the riparian zone changes with wetness state on P2L16:

"Moreover, wetness state changes the chemical function of the RZ (Vidon, 2017)."

5. It would seem fully screened wells down to apprx 1.5 m depth would integrate shallow 'DRIP' water and deeper mineral soil water, how did you account for this?

Answer: The wells were drilled until resistance or until a first aquitard was encountered. Given the exponentially decaying hydraulic conductivity profile, we can assume that the majority of the water is DRIP (or non-DRIP) water. We have addressed this on P5L2:

"Given that context, lateral flow below the well bottom was considered negligible compared to the flow in the vertical domain of our well installations."

I think your statements in the Discussion section regarding 'old' and 'young' water are a bit to speculative, as this is essentially only based on EC data, a parameter influenced by a number of flowpath process. It does not seem like any age dating/isotope analysis was performed, so how confident can you be regarding relative water ages/residence times?

Answer: We agree that our young/old water discussion is speculative, and we have replaced this part of the discussion.

Also, you mention the piezometers were installed until they reached a hard layer. Did this depth vary systematically from DRIP to non-DRIP locations? If so you could include that data, as depth to rock/confining layer can also be a strong control on shallow groundwater flowpath chemistry.

Answer: We have included well depth information on P5L19:

"All wells were drilled until resistance, or an aquitard layer. Riparian wells had a mean depth of 95 cm, transition wells 99 cm, and upland wells 121 cm."

Fig 1: Although we might expect local shallow percolation in non-DRIP near stream zones, groundwater flow is likely dominated by the lateral component toward the stream (in gaining stream reaches), though the discharge magnitude may be reduced compared to preferential discharge points. I suggest you alter the 'vertical groundwater flow' language in the 3rd panel of this Figure, the vertical flow you refer to may instead by non saturated percolation toward the water table, where groundwater flow is pre- dominantly horizontal.

Answer: We agree that non-saturated percolation is a more appropriate term for this process and we changed the figure accordingly

Have you measured any vertical head gradients at the wells, and lateral gradients between wells, to support these conceptual diagrams?

Answer: We have provided exemplar groundwater level data in Figure S1. Unfortunately we have no lateral gradients to further support our conceptual diagram.

Fig 3- The caption could be simplified, you do not need to define DRIPs in the caption as this is done in the text

Answer: We changed the caption

Minor points:

Pg 2:

L2 repeat of the word 'landscape', please look for replacement

Answer: We corrected this

L3 I am not sure what you mean by 'newly introduced water', can you be more specific?

Answer: This sentence changed to (P2L4): "Lateral groundwater inputs account for a large part of the streamflow of small streams, magnifying groundwater controls on stream CO2 emissions (Hotchkiss et al., 2015)."

L19 do you have a reference example to associate with: 'Traditionally, streamflow generation has often been assumed to be driven by spatially diffuse groundwater exchange often released at a constant rate.'?

Answer: This sentence and the following sentence changed to (P2L20):

"In hydrological models streamflow generation has often been conceptualized as a diffuse process, which limits the ability to express points of focused groundwater discharges (Briggs and Hare, 2018)."

**Pg 4**

L20: could cite here the hydrographs shown in Ploum et al 2018

**Answer: We added the reference**

L24: you would not consider the fall period to be 'baseflow' dominated as well or is this just a winter condition in your watershed system?

Answer: We considered baseflow as the flow conditions in winter given the snow dominated region we performed our study. In autumn low flow conditions can occur but not as long and consistent as during winter due to regular rain events. In winter snow and soil frost inhibit flow in the shallow subsurface and 'true' baseflow conditions occur.

Pg 7

L13: replace 'double as high'

Answer: We have changed this to 'twice as high'

Variability in landscapes is a challenge for understanding how landscapes influence water chemistry in space and time. By developing a sampling design based on a hypothesis about how water is flowing through the riparian zone, this study has provided new insights into the hydrobiogeochemical structures that shape the connection between landscapes and waters. This has practical implications for the design of buffer strips that are widely used in water management. As such this paper can be a valuable contribution to the literature. I think its value would be enhanced if a few points in the paper got further attention.

**Answer: Thank you for providing this short comment, we will incorporate the points in our revised manuscript.**

One concerns the distinction drawn between DRIPs and the confluences of ephemeral streams on page 5, lines 19-21. The text here was not clear. I think the authors are trying to say that such confluences are not included in their definition of DRIPS since DRIPS do not have clear channels. It would be good if this could be clarified.

Answer: We replaced this section and included a paragraph in the discussion to clarify the context of DRIPs in relation to ephemeral streams and other features that have been studied. Figure S2 was added to demonstrate how a DRIP looks like in the landscape. We agree that emphemeral stream and DRIPs differ from each other by the presence of a clear channel in the case of emphemeral streams, while this is not the case for DRIPs. In practice it can occur that DRIPs have a small channel-like appearance within the very last meter when it merges with the stream, for example when there is bank height difference.

A second point I suggest that the authors address concerns the discussion of relative contributions from DRIPs and Non-DRIPS to stream chemistry under different flow conditions (page 10, lines 29-34). This part of the discussion talks about the contrasting chemistries coming from DRIPS and non-DRIPS. But the effect of chemical differences in the source waters on stream chemistry depends on the proportion of water coming from the different source waters. Is there some assumption underlying this part of the discussion about how much water comes from DRIPs relative to non-DRIPS during high and low flow conditions? Clarification of that would help make the points in this part of the discussion more persuasive.

Answer: We have rewritten the discussion, and reduced the implications for streamflow. However we have added an example of the spring flood (P914) to demonstrate that rising groundwater levels in non-DRIP riparian zone results in higher DOC concentrations. However we have not further elaborated on the implications for stream chemistry here to remain the focus on groundwater related topics. As pointed out, the change in water sources of the stream during events, likely means that non-DRIPs play a larger role than during low flow conditions where groundwater levels are lower. We have added Figure S1 to demonstrate this. We have reported earlier that detectability of DRIPs changes throughout events and seasons (Ploum et al., 2018), which also suggests that DRIP/non-DRIP contributions vary over time.

Furthermore, if the DRIPS do not include the confluences of intermittent streams with the perennial stream channel, it would be important to mention what these ephemeral streams are doing to contribute to the high-flow stream chemistry being talked about in the discussion.

Answer: We agree that intermittent streams should be included when evaluating high-flow stream chemistry. A clean channel-like feature might respond faster to hydrological inputs compared to

DRIPs. That being said, relative to non-DRIP riparian hillslopes, the responsiveness of shallow subsurface runoff generation of DRIPs possibly falls in the same order of magnitude as intermittent streams. Similar to the comment above, we have reduced the speculations about stream implications. On P10L8 we highlight that ephemeral streams and DRIPs are two different features that both focus water towards perennial networks.

One final question, the concept of a "DRIP initiation threshold" is mentioned on page10 line 35, but a definition of what this means is not given. Please explain the term.

Answer: We have changed this section and elaborated on the selection process by providing the ranges of contributing areas. P5L9:

"The DRIPs (n = 10) were selected with contributing upslope area varying from 0.6 to 7.7 ha, with a mean contributing area of 2.7 ha. Non-DRIPs had an upslope contributing area between 4 and 80 m2 (on average 17 m2)."

**Are DOC concentrations in riparian groundwater linked to hydrological pathways in the boreal forest? Are hydrological pathways and variability in groundwater chemistry linked in the riparian boreal forest?**

**5 Stefan W. Ploum1, Hjalmar Laudon1, Andrés Peralta-Tapia2, Lenka Kuglerová1**

1Department of Forest Ecology and Management, Swedish University of Agricultural Sciences, 901 86 Umeå, Sweden 2Department of Ecology and Environmental Sciences, Umeå University, 901 87 Umeå, Sweden

Correspondence to: Stefan W. Ploum (stefan.ploum@slu.se)

**10**

Abstract.

[revised manuscript text omitted]

---

## Editor Decision (ED1)

General comment
The authors did a good job revising their manuscript. Mostly all of the points raised have been adequately addressed.

Author response: Thank you for reviewing the revised manuscript

However, I still am not convinced by the discussion chapter. The authors confirm their hypothesis but miss the chance to discuss why DRIPs and non-DRIPs are different in space and time. There is a short discussion on the spring season specifically but non on the seasonality (factor TIME) in general nor on the factor POS (position relative to the stream). Not discussing the potential processes behind observation weakens the paper. I thus strongly encourage the authors to use the discussion for that and not for the implications of their observations (which, to my opinion rather belongs into the concluding sections).

Author response: we agree with the reviewer that this part of the discussion can be improved. We have extended the second paragraph to discuss our own results related to space and time factors and more clearly interpret the different findings of DRIP and non-DRIPs in terms of space and time .

Specific comments
Abstract
L11: Would „can change groundwater chemistry" be more precise here?

Author response: we have adopted the suggestion
L25: "so-called" can be omitted

Author response: we removed "so-called"
L26: do you refer to homogeneity in groundwater quality / chemistry here? You may add this to be clearer.

Author response: We have changed the sentence to: "Moreover, groundwater chemistry from DRIPs was spatially and temporally homogeneous."

Introduction
P2 L16-17: "changes the chemical fuction of the RZ" – add "in time" here?

Author response: we have adopted the suggestion
P3 L3: "However" does not seem to fit here.

Author response: We changed the sentence to: "These Discrete Riparian Inflow Points provide…."
P3 L15: Yes, Kirchner 2003 assumes a mixed groundwater reservoir but does not specifically addresses/ studies groundwater quality but rather surface water age and chemistry. Does not seem to fit here.

Author response: We have removed this part of the sentence and added the following reference to further support the statement:

Tetzlaff, D. and Soulsby, C.: Sources of baseflow in larger catchments–Using tracers to develop a holistic understanding of runoff generation, Journal of Hydrology, 359(3-4), 287–302, 2008.

P3 L38: Consider starting a new section here.

Author response: adopted

P3 L38-44: Consider to broaden the objectives. You aim at proving a hypothesis. You could add that you also discuss implications in terms of…

Author response: we have incorporated the sentence:

"Furthermore we discuss the implications of using a binary categorization of the riparian zone opposed to continuous, process based approaches."

Fig. 1: Groundwater quality differences are a hypothesis so far. Can you make this clear in captions or in the figure itself? Is the "headwater lake" a common feature for boreal headwaters or specific for the studied catchment here?

Author response: we changed the first sentence of the caption to:

"Conceptual illustration of the two types of hypothesized riparian areas along a boreal stream."

The example is specific to the studied catchment, but in the boreal landscape it is common for headwaters to emerge from small lakes and mires.

Methods
P5L20: Add standard deviation or range to the mean depth.
P6 L3: Add standard deviation or CV (that makes variability best comparable between DRIPs and non-DRIPs) here as well.

Author response: we have added the standard deviations for both cases
P6 L11: Add the basic hydrological / hydroclimatic conditions – e.g. later on you talk about spring flood.

Author response: We have added the yearly Q and P for the years 2016 and 2017.

Results
Figure 3: Write in captions what "p" is indicating.

Author response: we extended the caption
Discussion
P9 L16-19: The described ice effect will create dilution in surface water but not in groundwater as touched on here, right? This is the only discussion of the results that I found in the entire discussion section.

Author response: The surface runoff, or ice runoff, will reach the groundwater sampling well since the well is fully screened from top to bottom. We clarified this in the sentence.
P10 L1-15: I am not sure if this section on the size of contributing area is well placed here.

Author response:  we agree that this is an abrupt change of topic, however the difference in contributing area is an important aspect of the DRIP/non-DRIP categorization. We think that moving this section either up or down would not suit the rest of the paragraphs.
P10 L 35 groundwater that is "incorporated in the stream" has an unclear meaning

P11 L1 -21: This section is on implications and not a discussion on the meaning of the results and should rather be moved to the 
[revised manuscript text omitted]

---

## Author Response (AR3)

Dear Editor,

Thank you for reviewing the revised manuscript and providing the technical corrections. We have adapted the manuscript accordingly and further checked for technical details and included a few clarifications. To clarify the second paragraph of the discussion we have expanded the section. The corrections can be found below highlighted in color.

Kind regards,

Stefan Ploum

[revised manuscript text omitted]
 2018. The upper panel shows six time series of water tables relative to the soil surface. Non-DRIPs are represented in orange, yellow and red dotted lines (well numbers 502, 504, 506). In the three shades of blue DRIP wells are demonstrated (wells 503, 505, 507). All wells were in a 5 meter distance from the stream, mostly within the first 2 meters. In the second panel specific discharge is presented over the same period. This is the discharge gained from the riparian zone, based on two gauging stations upstream (C5) and downstream (C6) of the stream reach.**

[Figure]

**Figure S2 Photograph of a DRIP in July 2017 by Stefan Ploum**

Preliminary analysis of DRIPs across the Krycklan catchment:

5    For the preliminary analysis of DRIP coverage across the Krycklan catchment the following approach was followed:

The stream network was defined by a 10 ha flow initiation threshold using a 2 meter DEM. Then a DRIP network was defined using a 2 ha initiation threshold. Each point where the 2 ha stream network was incorporated in the 10 ha network, was considered as a DRIP site. The area of the catchment was 62 km$^2$. The contributing area of the DRIPs was 35.34 km$^2$, which is 57 % of the catchment area. The total length of the stream network was 162.5 km. We considered the total length of both

10   sides of the stream as the riparian zone, which was 325 km. The total length of stream banks where DRIPs flow into the stream network was 20.75 km, when assuming a width of 25 meters for each DRIP (n=830). The total area of DRIPs was 12.8% of the total length of stream banks of the 10 ha stream network.